

# Local operator entanglement in spin chains

**Eric Mascot[1]⋆, Masahiro Nozaki[2,3] and Masaki Tezuka[4]**

**1** Department of Physics, University of Illinois at Chicago, Chicago, IL 60607, USA
**2** iTHEMS Program, RIKEN, Wako, Saitama 351-0198, Japan
**3** Kavli Institute for Theoretical Sciences and CAS Center for Excellence in Topological
Quantum Computation, University of Chinese Academy of Sciences, Beijing, 100190, China
**4** Department of Physics, Kyoto University, Kyoto 606-8502, Japan

⋆ eric.mascot@unimelb.edu.au

## Abstract

Understanding how and whether local perturbations can affect the entire quantum system is a fundamental step in understanding non-equilibrium phenomena such as thermalization. This knowledge of non-equilibrium phenomena is applicable for quantum computation, as many quantum computers employ non-equilibrium processes for computations. In this paper, we investigate the evolution of bi- and tripartite operator mutual information of the time-evolution operator and the Pauli spin operators in the one-dimensional Ising model with magnetic field and the disordered Heisenberg model to study the properties of quantum circuits. In the Ising model, the early-time evolution qualitatively follows an effective light cone picture, and the late-time value is well described by Page's value for a random pure state. In the Heisenberg model with strong disorder, we find that many-body localization prevents the information from propagating and being delocalized. We also find an effective Ising Hamiltonian that describes the time evolution of bi- and tripartite operator mutual information for the Heisenberg model in the large disorder regime.

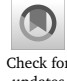

# 1 Introduction and summary

**Introduction**

Thermalization is one of the frontier research topics in theoretical and experimental physics, in which the entanglement of the state plays an important role [1–3]. In closed quantum systems, the unitarity of the time evolution operator prevents a pure state from becoming a thermal state. However, in many cases, after an adequate amount of time, observables in a subsystem are well-approximated by those for a thermal state, known as thermalization. Though the thermal state does not depend on the entanglement structure of an initial state, thermalization *does* depend on the initial state and the details of its dynamics. Here, the entanglement structure refers to the way subsystems are entangled with each other. The reduced density matrix for a subsystem is given by a mixed state. If time evolution washes out the dependence of a mixed state on the details of the initial state, the reduced density matrix becomes the reduced thermal state. An effective temperature of the thermal state is given by the energy scale of the initial state. Quantum entanglement has also been found to shed light on the mechanism behind the anti-de Sitter / conformal field theory (AdS / CFT) correspondence [4–6].

To better understand thermalization, authors in [7–10] studied the entanglement entropy in a quantum quench of conformal field theories. The entanglement entropy is the von Neumann entropy of the reduced density matrix. A reduced density matrix becomes approximately equal to the thermal state after enough time passes for the dynamics to wash out the information about the initial state.

The authors in [7] studied the time evolution of entanglement entropy for a subsystem of width $l$ in 1+1 dimensional (1+1D) CFTs. A given initial state depends on an initial energy scale, $1/\beta$. The time evolution of entanglement entropy after a quench follows the relativistic propagation of quasiparticles that are created by the quench. An entangled pair of particles is created on each site. Thereafter, one of the particles moves leftward at the speed of light, while the other moves rightward. When only one of these two particles remains in the subsystem, the entanglement entropy increases, owing to the entangled pair. For the early-time window, $\beta \ll t < \frac{l}{2}$, the entanglement entropy is a linear function of $t$, but when $\frac{l}{2} < t$, it is proportional to $l$. Thus, this volume law of entanglement entropy indicates that the subsystem has thermalized.

The authors in [11, 12] found that the time evolution of mutual information and entanglement entropy for two disjoint intervals in holographic CFT, a CFT that has a gravity dual, does not follow the quasiparticle interpretation. The early-time evolution follows the relativistic propagation of quasiparticles, but the late-time evolution does not. The authors in [13] found that holographic tripartite mutual information, a linear combination of holographic mutual information, must be non-positive. These articles indicate that the *multipartite entanglement* in CFTs depends on the details of the theories, such as the operator content. We will demonstrate how multipartite entanglement sheds light on the details of thermalization. In [14–24], it was found that the operator entanglement, the entanglement of state dual to the operator, elegantly captures the details of dynamics in non-equilibrium processes. The operator entanglement of the time-evolution operator is able to directly show that the quantum information in the initial subsystem is delocalized in the late-time regime, so that local observers are not able to obtain it.

After thermalization is complete, the entanglement entropy is expected to follow the Page curve [25, 26]. When the late-time entanglement entropy follows the Page curve, non-local correlation between the small subsystems vanishes. Quantum computation, a cutting-edge research field, can be implemented as a nonequilibrium process. In the quantum circuit model, quantum computations are constructed as a sequence of quantum gates and measurements. The quantum gates are local unitary operators and these local gates and measurements are arranged spatially [27, 28]. If we interpret each step of the computation as each time of the non-equilibrium process, the quantum circuit may be equivalent to the (non-unitary) non-equilibrium process. It is also important to understand quantum computations or nonequilibrium processes that preserve the quantum nature of the state. In this paper, we examine the properties of a quantum computational process by studying the operator entanglement in the two different dynamical systems. One system has dynamics with the strong scrambling ability and few degrees of freedom. The other has dynamics that preserve the quantum nature of the state. This is the spin chain in the many-body localization phase (MBL phase).

In this paper, we will study the properties of a quantum circuit with 1) strong scrambling ability, and 2) weak scrambling effect. We will study whether the late-time evolution of the bi- and tripartite operator mutual information (BOMI and TOMI) of local operators depends on the operators in the dynamical system. The TOMI measures how much information regarding the subsystem may be locally hidden, while the BOMI measures how many EPR pairs associated with the subsystem considered may be destroyed during the nonequilibrium processes. For 1) strong scrambling, the fact that the late time evolution of BOMI and TOMI are independent of the local operator suggests that the information of the local operator is washed out by the dynamics. The authors in [23] found that in the system with strong scrambling ability and large degrees of freedom, the late-time behavior of BOMI and TOMI of local operators is independent of the operators and approximated by that of the unitary operator. Dynamics with a large number of degrees of freedom and strong scrambling has several different features from dynamics with only a small number of degrees of freedom. For example, the out of time ordered correlator (OTOC) can measures such properties of dynamics that depend on the number of degrees of freedom [29, 30]. For this reason, even if the degrees of freedom are small, we will study whether the late time behavior of BOMI and TOMI depend on local operators. We will also study how the finiteness of the system affects the scrambling effect of the dynamics. For comparison, we consider the disordered spin chain, which is expected to be a remarkable example of dynamics with weak scrambling effects. This spin system has chaotic and MBL phases corresponding to weak and strong disorder, respectively. By varying the strength of the disorder, we will study the difference in the evolution of BOMI and TOMI.

## Summary of results

We investigate the time evolution of bi- and tripartite operator mutual information of the time-evolution operator and Pauli's spin operators for three configurations, the fully-overlapping, partially-overlapping, and disjoint configurations, (See Figure 2) in a spin chain with strong scrambling ability and a disordered spin chain.

We divide the input (original) Hilbert space into $A$ and $\bar{A}$, and divide the output (scattered) Hilbert space into $B$ and $\bar{B}$. In the fully- or partially-overlapping configurations, $A$ is initially in $B$. In the disjoint configuration, $A$ is out of $B$. We summarize the results here.

### Spin chain with strong scrambling ability

We consider a 1D Ising model with transverse and longitudinal field for its strong scrambling ability. We find that the time-evolution of bi- and tripartite operator mutual information is described by two effective descriptions, an effective light cone and the Page curve.

**Early time evolution:** We find the early-time evolution of BOMI and TOMI depend on the various operators and boundary conditions, and is described qualitatively by an effective light cone picture. In this picture, while sections of $A$, $\bar{A}$, $B$ and $\bar{B}$ are partially inside the light cone, the bipartite mutual information decreases.

**Late time evolution:** We find the late-time evolution of BOMI and TOMI is independent of various operators and boundary conditions, and the late time value is described by Page's value for the average entropy [25,26], which we call the Page curve (See, Section 3.2). This suggests that the strong scrambling effect washes out the information about the local operators and boundary conditions, so that the state becomes the typical one, the state which is independent of the initial state. We also find that the late time value of BOMI and TOMI can depend on the system size. This suggests that the finiteness of the system can prevent the reduced density matrix for $A \cup B$ from factorizing into the reduced density matrices for $A$ and that for $B$, and the finiteness of the system can also reduce the amount of information scrambled by the the dynamics.

### Disordered spin chain

We consider a 1D Heisenberg model with uniform disorder strength to study the many-body localization phase. We find parallels to the simpler Ising chain for the weak and strong disorder regimes.

**Weak disorder:** The spin chain with weak disorder is in the chaotic phase. Therefore, the late-time values of BOMI and TOMI is well-described by the Page curve. Also, in the weak disorder region of the disordered spin system, information on local operators and boundary conditions is lost locally due to dynamics.

**Strong disorder:** When disorder is large, the late-time value of BOMI in the full overlap configurations increases with the strength of disorder. We interpret this to be due to the localization effect of the system. When the disorder is large enough, BOMI and TOMI periodically evolve in time. Thus, BOMI and TOMI exhibit quantum revival. This oscillation is well-described by an effective Hamiltonian. The period is determined by the strength of local interaction, and the periodic behavior is due to the off-diagonal components of the reduced density matrix in the $\sigma_z$ basis. This suggests this dynamical system is the quantum circuit that preserves the quantum nature of the initial state.

The authors in [24] have studied the late-time dynamics of many body localized systems by using operator mutual information and other quantities such as OTOC. Our results are consistent with theirs and we extend their results on operator mutual information to early-time dynamics.

## 2 Preliminary

In this section, we first describe the concepts of information scrambling and operator spreading. Second, we review many-body localization, which is an interesting dynamical phenomenon far from chaotic dynamics. Finally, we introduce operator entanglement and bipartite and tripartite operator mutual information as quantum measures for operator entanglement.

### 2.1 Information scrambling and operator spreading

Let us briefly review information scrambling and operator spreading.

**Information scrambling**

Information scrambling occurs in dynamical processes where the subsystem thermalizes after an adequate amount of time. To explain information scrambling, we first review thermalization of subsystems. Consider the initial state

$$|\Psi\rangle = \sum_{E_\alpha < 1/\epsilon} C_\alpha |\alpha\rangle \,, \tag{1}$$

where $|\alpha\rangle$ is an eigenstate of Hamiltonian, $H|\alpha\rangle = E_\alpha |\alpha\rangle$ and $E_\alpha \geq 0$. his initial state is characterized by an energy scale, $1/\epsilon$.

Following the time evolution $U(t)$, the reduced density matrix for a subsystem $\mathcal{V}$ is given by

$$\rho_\mathcal{V}(t) = \mathrm{Tr}_{\bar{\mathcal{V}}}\left[U(t)|\Psi\rangle\langle\Psi|U^\dagger(t)\right], \tag{2}$$

where we divide the total Hilbert space into $\mathcal{V}$ and $\bar{\mathcal{V}}$. The reduced density matrix $\rho_\mathcal{V}(t)$ depends not only on the initial state, but also on the details of time-evolution operator $U(t)$. In this paper, the definition of subsystem thermalization is that $\rho_\mathcal{V}(t)$ is well-approximated by a reduced thermal density with an effective inverse temperature $\beta_{\mathrm{eff}} \approx \epsilon$ after some time,

$$\rho_\mathcal{V}(t) \xrightarrow{t \to \infty} \rho_{\mathrm{Th}.\mathcal{V}} = \frac{\mathrm{Tr}_{\bar{\mathcal{V}}} e^{-\epsilon H}}{\mathrm{Tr}\, e^{-\epsilon H}} \,, \tag{3}$$

where $\rho_{\mathrm{Th}.}$ is the thermal state. If the late time behavior of observables in $\mathcal{V}$ are approximated by their thermal expectation value, then we say that thermalization of $\mathcal{V}$ occurs. For example, in addition to the mutual information content studied in this paper, the one-point function and the distance between states [31–33] are also used as indicators of thermalization of subsystems. If the subsystem thermalizes, these observables behave as follows:

$$\lim_{t \to \infty} \frac{\langle\Psi|U^\dagger(t)V(x)U(t)|\Psi\rangle}{\langle V(x)\rangle_{\mathrm{Th}.}} \approx 1 \,, \tag{4}$$

$$\lim_{t \to \infty} \frac{\mathrm{Tr}_\mathcal{V}\left(\rho_\mathcal{V}(t)\mathrm{Tr}_{\bar{\mathcal{V}}} e^{-\epsilon H}\right)}{\sqrt{\mathrm{Tr}_\mathcal{V}\left(\mathrm{Tr}_{\bar{\mathcal{V}}} e^{-\epsilon H}\right)^2 \mathrm{Tr}_\mathcal{V}\left(\rho_\mathcal{V}(t)\right)^2}} \approx 1 \,, \tag{5}$$

where, $\langle \cdot \rangle_{\text{Th.}}$ denotes the expectation value of the thermal state. If the approximation in (3) works well, the late-time reduced density matrix depends only on the initial energy $1/\epsilon$ and not on the entanglement structure of its initial state. Information scrambling is the phenomenon where the late-time reduced density matrices for any subsystem is independent of the structure of its initial state, and instead depends on $1/\epsilon$. If this phenomenon occurs, dynamics has the strongest scrambling ability.

**Operator spreading**

One interesting dynamical phenomenon is operator spreading [14, 34–36]. A local operator in the Heisenberg picture $\mathcal{O}(x, t)$ is given by

$$\mathcal{O}(x, t) = e^{iHt} \mathcal{O}(x) e^{-iHt}$$
$$= \mathcal{O}(x) + (it)[H, \mathcal{O}(x)] + \frac{(it)^2}{2!}[H, [H, \mathcal{O}(x)]] + \cdots, \tag{6}$$

where we used the Baker-Campbell-Hausdorff formula. If $t$ is small, $\mathcal{O}(x, t)$ is approximately given by a simple operator $\mathcal{O}(x)$. As time proceeds, the larger the contributions of terms with higher powers of $t$ become. Thus, a local operator in the Heisenberg picuture $\mathcal{O}(x, t)$ becomes more complicated over time.

Consider a probe $\mathcal{Q}(y)$, a local operator far from the position of $\mathcal{O}$, to measure how much $\mathcal{O}(x, -t)$ spreads. If $t$ is small, this probe is causally unrelated to $\mathcal{O}(x, -t)$,

$$[\mathcal{Q}(y), \mathcal{O}(x, -t)] = 0. \tag{7}$$

If $t$ is large, this probe is causally related to $\mathcal{O}(x, -t)$,

$$[\mathcal{Q}(y), \mathcal{O}(x, -t)] \neq 0. \tag{8}$$

This is *operator spreading*. The expectation value of the square of the commutator $\langle [\mathcal{Q}(y), \mathcal{O}(x, -t)]^2 \rangle$ measures how much $\mathcal{O}(x, -t)$ spreads and how complicated it becomes. The behavior of this expectation value of the commutator square is related to the out-of-time-ordered correlator (OTOC), which has been actively studied [18, 29, 37–39]. The OTOC is defined as

$$C(t) = \left\langle \mathcal{O}^\dagger(x, -t) \mathcal{Q}^\dagger(y) \mathcal{O}(x, -t) \mathcal{Q}(y) \right\rangle, \tag{9}$$

where $\langle \cdot \rangle$ denotes the thermal expectation value with inverse temperature $\beta$. The authors in [37] suggest that the form of OTOC in the large $N$ theories for the time region $t \gg \beta$ is given by

$$C(t) \underset{t \gg \beta}{\approx} c_0 - \frac{c_1}{N^2} e^{\lambda t}, \tag{10}$$

where $c_0$ and $c_1$ are positive, order one, quantities that depend on $\mathcal{O}$ and $\mathcal{Q}$, and $\lambda$ is the Lyapunov exponent, which depends on the scrambling effect of the dynamics. In holographic CFTs, $\lambda$ is maximized at $\frac{2\pi}{\beta}$. In generic systems, including the spin chain considered in this paper, the behavior of OTOC is discussed in [40].

## 2.2 Many-body localization (MBL)

Many-body localization is a counterexample to thermalization, which says the structure of quantum entanglement is almost preserved under time-evolution. Therefore, we expect quantum information to be preserved locally.

The localization of energy eigenstates up to finite energy density in interacting quantum systems, termed many-body localization, has received wide attention for more than a decade. See [41–43] for recent review articles.

In the non-interacting case the single-particle orbitals can localize due to randomness [44]. It was initially not clear whether localization persists in the presence of interactions. In the tight-binding picture, a particle localizing around a lower energy lattice site would repel other particles if the interaction between the particles is repulsive, as would be expected for electrons with Coulomb interactions. Then the effect of the interaction would be to weaken the randomness in the site energy. In later studies, the possibility of localization was first discussed for the many-body ground state [45–47], and then to finite-temperature cases [48–50]. Interacting systems staying insulating at finite temperatures are said to be many-body localized.

An MBL state can escape thermalization when placed in a non-equilibrium initial state. Integrable systems also do not generally thermalize, but the integrability sensitively depends on the choice of parameters and thermalization would happen once the integrability is broken, unless MBL is established. MBL phases are robust against weak perturbations since all the energy eigenstates are localized up to a finite energy density. In systems with a set of random parameters (e.g. hoppings, site energies, or on-site interactions) drawn from some probability distributions, any physical quantities would be probabilistically determined. However, once the constants controlling these distributions satisfy certain conditions, the probability that the MBL behavior is not observed is exponentially small, then the system controlled by these constants can be said to be in the MBL phase.

Note that systems that exhibit MBL without randomness in the Hamiltonian have been proposed [51]. Also, note that systems with spatially quasiperiodic modulation, whose non-uniform feature is determined by a few parameters as opposed to lacking long-range correlation, can be many-body localized. Even in these cases, we can generally expect the many-body localized phase to be robust against time-independent weak perturbations.

Hamiltonians exhibiting the MBL phase provide counterexamples to the most general, or naive, form of the eigenstate thermalization hypothesis (ETH) [1, 2]. ETH states that for an isolated quantum mechanical many-body system, for any initial state given as a linear combination of eigenstates close in energy, the expectation value of an operator should "thermalize", that is, relax to a value close to the one expected for the microcanonical ensemble determined by the energy range and will only show small fluctuations at later times.

A kind of emergent integrability exists in the many-body localized systems, as there are exponentially many quasi-local integrables of motion, often abbreviated as LIOMs. Correspondingly, the energy spectrum does not show the random matrix universality as expected in a quantum chaotic system [52] and become uncorrelated.

The time evolution of entanglement entropy from an uncorrelated state, which would be linear in non-interacting systems or general non-integrable systems without localization [53], becomes logarithmic in time for a many-body localized system [54–57]. In a many-body localized eigenstate the entanglement entropy for a subsystem exhibits area law dependence on the surface area rather than volume laws that is expected for a thermal reduced density matrix for the subsystem obeying ETH [58].

For a class of one-dimensional spin chain with random local *interactions*, the existence of MBL has been proved. [59]. On the other hand, the best studied model is the one-dimensional chain of $S = 1/2$ spins with random *fields*, eq. (34). While recent studies have suggested that the MBL transition does not occur at $w/J \approx 3$ as was previously believed, but occurs at $w/J \gtrsim 20$ or at a large $w/J$ increasing as the system-size is increased [60,61], for system sizes that allow exact diagonalization studies, the phenomenology of the many-body eigenstates do exhibit localizing behavior for $w/J \approx 3$.

Experimentally, MBL has been observed in ultracold atoms [62–65] and ions [66] trapped in low dimensions. In [62], the atoms were prepared in a highly non-equilibrium density-wave state, in which most of the particle population is on every second lattice site in a one-dimensional lattice. Then, the system was allowed to evolve under the given Hamiltonian

and the imbalance between the even and odd sites was measured. In the presence of the on-site interaction, the imbalance remained finite when the quasiperiodic potential was strong enough.

## 2.3 Operator entanglement and its quantum measures

In this section, we describe operator entanglement and the quantum measures considered in this paper, bipartite operator mutual information (BOMI) and tripartite operator mutual information (TOMI).

### 2.3.1 Definition of operator entanglement

Operator entanglement is defined as the entanglement structure of the dual state, which is the state given by a state-channel map:

$$\mathcal{W} = \sum_{i,j} \mathcal{W}_{ij} |i\rangle \langle j| \xrightarrow{\text{state-channel map}} |W\rangle = \mathcal{N} \sum_{i,j} \mathcal{W}_{ij} |i\rangle |j^*\rangle, \tag{11}$$

where $\mathcal{W}$ is an operator, $|\cdot^*\rangle$ is a $CPT$ conjugate of $|\cdot\rangle$, and $\mathcal{N}$ is a normalization factor. The dual state lives in the Hilbert space whose dimension is the square of the dimension of the original Hilbert space:

$$\mathcal{H}_{\mathcal{W}} \xrightarrow{\text{state-channel map}} \mathcal{H}_{|\mathcal{W}\rangle} = \mathcal{H}_{\mathcal{W}} \otimes \mathcal{H}_{\mathcal{W}}, \tag{12}$$

where $\mathcal{H}_{\mathcal{W}}$ is the Hilbert space where the operator acts, and $H_{|\mathcal{W}\rangle}$ is the space where the dual state lives.

In this paper, we consider the operator entanglement of the following operators.

**Unitary operator**

Time evolution changes the entanglement structure of an initial state. The quantum correlation between a subsystem $A$ of an initial state and subsystem $B$ of the time-evolved state should show how the time evolution changes the local structure of an initial state. The information about the local structure propagates (and/or is delocalized) as the structure changes. In this paper, we define operator entanglement measurements of the unitary time-evolution operator, and we study *the spreading and delocalization of quantum information* by using the measures for operator entanglement defined in Section 2.3.3. The unitary channel for the time-evolution operator $U(t)$, represented in the eigenbasis of the Hamiltonian, is

$$U(t) = e^{-iHt} = \sum_{a}^{N} e^{-itE_a} |a\rangle \langle a|, \tag{13}$$

where the Hamiltonian is a time-independent operator, $|a\rangle$ is an eigenstate, and $N$ is the number of states.

The dual state to (13) can be defined by the state-channel map $f[U(t)]$,

$$f[U(t)] = |U(t)\rangle = \frac{1}{\sqrt{N}} \sum_{a}^{N} e^{-itE_a} |a\rangle |a\rangle = \frac{1}{\sqrt{N}} \sum_{a}^{N} e^{-itE_a} |a\rangle |a\rangle$$

$$= \frac{1}{\sqrt{N}} \sum_{a}^{N} e^{-\frac{it}{2}(H_{\text{in}}+H_{\text{out}})} |a\rangle_{\text{in}} |a\rangle_{\text{out}}, \tag{14}$$

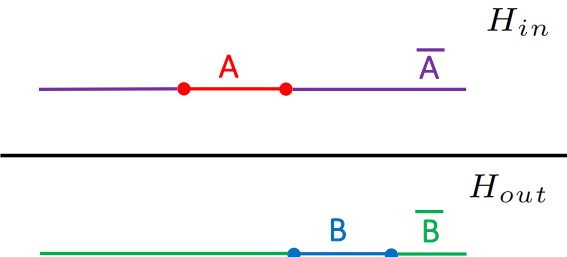

Figure 1: Input ($A$) and output ($B$) subsystems in the doubled Hilbert space $H_{\text{in}} \otimes H_{\text{out}}$. We compute the BOMI and TOMI for the configurations corresponding to Figure 2.

where we take $|a\rangle$ to be a real function because $|a\rangle$ is an eigenstate of $H$. The states, $|\cdot\rangle_{\text{in}}$ and $|\cdot\rangle_{\text{out}}$, are in the Hilbert spaces, $\mathcal{H}_{\text{in}}$ and $\mathcal{H}_{\text{out}}$. Thus, the state defined in (14) is in $\mathcal{H} = \mathcal{H}_{\text{in}} \otimes \mathcal{H}_{\text{out}}$. Its density matrix is given by

$$\rho = |U(t)\rangle\langle U(t)| = \frac{1}{N}\sum_{a,b=1}^{N} e^{-it(E_a - E_b)}|a\rangle_{\text{in}}\langle b|_{\text{in}} \otimes |a\rangle_{\text{out}}\langle b|_{\text{out}}. \tag{15}$$

We divide the input Hilbert space $\mathcal{H}_{\text{in}}$ into $\mathcal{H}_A$ and $\mathcal{H}_{\bar{A}}$ and the output Hilbert space $\mathcal{H}_{\text{out}}$ into $\mathcal{H}_B$ and $\mathcal{H}_{\bar{B}}$ as in Figure 1. The input and output states in the spacetime language are given by

$$|a\rangle_{\text{in}} = \sum_{iI} C_{iI}^a |i\rangle_A |I\rangle_{\bar{A}}, |b\rangle_{\text{in}} = \sum_{jJ} C_{jJ}^b |j\rangle_A |J\rangle_{\bar{A}},$$

$$|a\rangle_{\text{out}} = \sum_{\alpha\mathcal{A}} D_{\alpha\mathcal{A}}^a |\alpha\rangle_B |\mathcal{A}\rangle_{\bar{B}}, |b\rangle_{\text{out}} = \sum_{\beta\mathcal{B}} D_{\beta\mathcal{B}}^b |\beta\rangle_B |\mathcal{B}\rangle_{\bar{B}}, \tag{16}$$

where $|x\rangle_X$ are orthogonal vectors in $\mathcal{H}_X$ and $X = A, \bar{A}, B, \bar{B}$. Thus, the density matrix in the spacetime language is given by

$$\rho = \frac{1}{N}\sum_{a,b,i,j,I,J,\alpha,\beta,\mathcal{A},\mathcal{B}} e^{-it(E_a - E_b)} C_{iI}^a C_{jJ}^{b*} D_{\alpha\mathcal{A}}^a D_{\beta\mathcal{B}}^{b*} |i\rangle\langle j|_A \otimes |I\rangle\langle J|_{\bar{A}} \otimes |\alpha\rangle\langle\beta|_B \otimes |\mathcal{A}\rangle\langle\mathcal{B}|_{\bar{B}}. \tag{17}$$

Next, we consider a reduced density matrix for $A \cup B$ as an example. The matrix $\rho_{A\cup B}$ is given by limiting the sum in the above to $I = J$ and $\mathcal{A} = \mathcal{B}$,

$$\rho_{A\cup B} = \text{Tr}_{\overline{A\cup B}}\rho = \frac{1}{N}\sum_{a,b,i,j,J,\alpha,\beta,\mathcal{B}} e^{-it(E_a - E_b)} C_{iJ}^a C_{jJ}^{b*} D_{\alpha\mathcal{B}}^a D_{\beta\mathcal{B}}^{b*} |i\rangle\langle j|_A \otimes |\alpha\rangle\langle\beta|_B$$

$$= \sum_{a,b,i,j,J,\alpha,\beta,\mathcal{B}} e^{-it(E_a - E_b)} \mathcal{M}_{ij}^{ab} \mathcal{N}_{\alpha\beta}^{ab} |i\rangle\langle j|_A \otimes |\alpha\rangle\langle\beta|_B, \tag{18}$$

where $\mathcal{M}_{ij}^{ab} = \sum_J C_{iJ}^a C_{jJ}^{b*}$ and $\mathcal{N}_{ij}^{ab} = \sum_\mathcal{B} D_{\alpha\mathcal{B}}^a D_{\beta\mathcal{B}}^{b*}$. Operator entanglement entropy (OEE) for $\rho_{A\cup B}$ is defined as the von Neumann entropy of the reduced density matrix

$$S_{A\cup B} = -\text{Tr}_{A\cup B}\rho_{A\cup B} \log_2 \rho_{A\cup B} = -\sum_{\lambda_{A\cup B}} \lambda_{A\cup B} \log_2 \lambda_{A\cup B}, \tag{19}$$

where $\lambda_{A\cup B}$ are the eigenvalues of $\rho_{A\cup B}$.

By computing the mutual information (defined in Section 2.3.3) of the unitary time-evolution operator, we study the correlation between a subsystem of the initial state and a subsystem of the output state in which we measure local observables.

**Local operator**

A local perturbation, a perturbation due to a local operator, spreads under the time evolution as in Section 2.1, so that the entanglement structure of the scattered state, the state which is acted with the local operator, should differ from the original state. By measuring the correlation between the subsystem $A$ of the original state and $B$ of the scattered state, we should find where information about the entanglement structure in $A$ propagates and how much the operator delocalizes information. Let us define the local operator entanglement as the quantum entanglement of local operator $\mathcal{O}(x, t)$ as follows.

A local operator in Heisenberg picture $\mathcal{O}(x, t)$ is defined as

$$\mathcal{O}(x, t) = e^{iHt} \mathcal{O}(x) e^{-iHt}, \tag{20}$$

where we assume that the Hamiltonian is time-independent. We define the state dual to (20) by the channel-state map:

$$|\mathcal{O}(x, t)\rangle = \mathcal{N} \sum_{a,b} e^{itE_a} \langle a|_{\text{out}} \mathcal{O}(x) |b\rangle_{\text{in}} e^{-itE_b} |a\rangle_{\text{out}} |b\rangle_{\text{in}} = \mathcal{N} \sum_{a,b} \langle a|_{\text{out}} \mathcal{O}(x, t) |b\rangle_{\text{in}} |a\rangle_{\text{out}} |b\rangle_{\text{in}}, \tag{21}$$

where the states $|\cdot\rangle_{\text{in}}$ and $|\cdot\rangle_{\text{out}}$ are the eigenstates of the time-independent Hamiltonian, and the normalization factor $\mathcal{N}$ is given by

$$\mathcal{N} = Z^{-\frac{1}{2}}, \qquad Z = \text{Tr}\left(\mathcal{O}^{\dagger}(x)\mathcal{O}(x)\right). \tag{22}$$

The entanglement structure of the state in (21) strongly depends on the Heisenberg operator. For example, if $\mathcal{O}(x, t)$ is the identity operator, then the entanglement structure of the state in (21) is the same as that of maximally entangled state. In the general case, $\mathcal{O}(x, t)$ can make the quantum entanglement structure of the state in (21) less entangled than that of the maximally entangled state. By computing the mutual information (defined in Section 2.3.3) of local operators, we study the correlation between subsystem $A$ of the original state and subsystem $B$ of the scattered state.

### 2.3.2 Initial state

Here, we consider the dual states at $t = 0$. At $t = 0$, the dual states to the unitary and Pauli's operators in Heisenberg picture are respectively given by

$$|U(t = 0)\rangle = \frac{1}{2^{\frac{L}{2}}} \sum_a |a\rangle_{\text{out}} |a\rangle_{\text{in}}, \quad |\sigma^{(i)}_{\alpha=x,y,z}(t=0)\rangle = \frac{1}{2^{\frac{L}{2}}} \sum_a \sigma^{(i)}_{\alpha=x,y,z} |a\rangle_{\text{out}} |a\rangle_{\text{in}}, \tag{23}$$

where $\sigma^{(i)}_{\alpha=x,y,z}$ acts on $i$-th site of $\mathcal{H}_1$. Thus, $|U(t = 0)\rangle$ is equal to the thermofield double state with zero inverse temperature, and $|\sigma^{(i)}_{\alpha=x,y,z}(t=0)\rangle$ is given by acting the Pauli's operator on the themofield double state. In terms of the eigenstates of $\sigma_z$ on each site, $|U(t = 0)\rangle$ and $|\sigma^{(i)}_{\alpha=x,y,z}(t=0)\rangle$ are given by

$$|U(t = 0)\rangle = \prod_{i=1}^{L} |\text{EPR}\rangle_i, \qquad |\sigma^{(i)}_{\alpha=x,y,z}(t=0)\rangle = \sigma^{(i)}_{\alpha=x,y,z} \prod_{j=1}^{L} |\text{EPR}\rangle_j, \tag{24}$$

where $|\text{EPR}\rangle_i$ is defined by $|\text{EPR}\rangle_i = \frac{1}{2^{\frac{1}{2}}} \sum_{\sigma^{(i)}_z = \uparrow, \downarrow} |\sigma^{(i)}_z\rangle_{\text{out}} |\sigma^{(i)}_z\rangle_{\text{in}}$. Here, $i$ means the $i$-th site of the system.

### 2.3.3 Quantum measures

Operator mutual information measures non-local correlation between subsystems. As mentioned in the previous sections, we investigate the time evolution of the operator mutual information in various configurations in order to find how operator spreading occurs under time-evolution for a system with strong scrambling ability. We also study the operator mutual information in a system which has a many-body localized phase.

**Bipartite operator mutual information (BOMI)**

Bipartite operator mutual information (BOMI) is defined as the linear combination of OEE

$$I_{A,B} = S_A + S_B - S_{A\cup B}\,, \tag{25}$$

where $S_A$ and $S_B$ are OEEs for the input (original) subsystem $A$ and the output (scattered) subsystem $B$. The last term in the right-hand side in (25) is OEE for $A\cup B$. The input and output subsystems are defined as the subsystems of $H_{\text{in}}\otimes H_{\text{out}}$ where the dual state to $U(t)$ lives, while the original and scattered subsystems are defined as the subsystems of $H_{\text{original}}\otimes H_{\text{scattered}}$ where the state dual to $\mathcal{O}(t,x)$ lives. The BOMI measures the correlation between the input (original) and output (scattered) subsystems. If $I_{A,B}$ vanishes, there are no correlations between $A$ and $B$, which means that the reduced density matrices of the subsystems are unrelated to each other.

**Tripartite operator mutual information (TOMI)**

Tripartite operator mutual information (TOMI) is defined as a linear combination of the BOMI as follows. We divide the output or scattered space into the subsystems $B$ and $\bar{B}$, and also divide the input or original space into $A$ and $\bar{A}$. The TOMI which we consider is given by

$$I_{A,B,\bar{B}} = I_{A,B} + I_{A,\bar{B}} - I_{A,B\cup\bar{B}}\,. \tag{26}$$

As in [13,14,21,22], the TOMI in (26) measures how much information regarding $A$ is delocalized over time. We will briefly explain why TOMI in (26) can be an indicator of the information scrambling. Suppose the initial state is given by the product of EPR states in (24). Suppose a subsystem $A\cup B$ is given by the union of a subsystem $A$ on $\mathcal{H}_1$ and $B$ on $\mathcal{H}_2$. Let $n_\alpha$ denote the number of EPR pairs included in the subsystem $\alpha$. For this initial state, the value of $I_{A,B}$ and $I_{A,\bar{B}}$ is proportional to $n_{A\cup B}$ and $n_{A\cup\bar{B}}$, respectively, while the value of $I_{A,B\cup\bar{B}}$ is proportional to $n_{A\cup B\cup\bar{B}}$. Since $n_{A\cup B} + n_{A\cup\bar{B}} = n_{A\cup B\cup\bar{B}}$, the value of TOMI is zero. In this paper, we use TOMI to study how many these EPR pairs are preserved in non-equilibrium processes. Under dynamics with scrambling ability, the information is locally hidden, so that $n_{A\cup B}$ and $n_{A\cup\bar{B}}$ deceases in time. Decrease of $n_{A\cup B}$ and $n_{A\cup\bar{B}}$ depends on dynamics. In this paper, we use TOMI to study how many of these EPR pairs are preserved in a non-equilibrium processes. In the time evolution induced by the time evolution operator with no scrambling ability, the value of TOMI is independent of time, and it is zero, while in the time evolution induced by the time evolution operator with stronger scrambling ability, the TOMI will have a larger negative value.

**Configurations**

As shown in Figure 2, we study the time evolution of the operator mutual information in three configurations: (a)full overlap, (b)partial overlap, and (c)disjoint configurations. In the full overlap configuration, the size of $A$ and $B$ are equal, $l_A = l_B$, and the edges of each subsystem are aligned. In the partial overlap configuration, the size of $B$ is larger than that of $A$ by $s$, $l_B = l_A + s$. The left edge of subsystems $A$ and $B$ are aligned. In the disjoint configuration, the subsystems $A$ and $B$ do not have any overlapping regions and the distance between the right

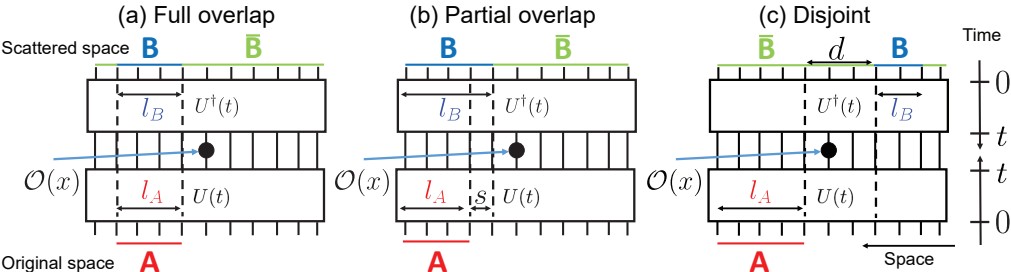

Figure 2: Schematic of the three configurations: (a) full overlap, (b) partial overlap, and (c) disjoint. Full overlap: The size of $A$ and $B$ are equal, $l_A = l_B$, and are aligned. Partial overlap: The size of $B$ is greater than $A$ by $s$, $l_A + s = l_B$, and are aligned on the left. Disjoint: The distance between the right boundary of $A$ and the left boundary of $B$ is $d$.

edge of $A$ and left edge of $B$ is $d$. These three configurations are useful to consider since in 2D CFTs [21, 23], how the dynamics affects the time evolution of BOMI depends on how the subsystems are taken. In the full overlap configuration, the behavior in time of BOMI does not depend as much on the scrambling ability of the time evolution operator. In the partial overlap and disjoint configurations, the time evolution of BOMI strongly depends on the scrambling ability of the dynamics. Therefore, we will study the time evolution of BOMI in these three configurations.

- **Full overlap configuration (Fig. 2 (a))** In this configuration, the initial value of $I_{A,B}$ for $|U(t)\rangle$ is proportional to the number of EPR pairs, both edges of which are in $A \cup B$. The decreases in time of $I_{A,B}$ will indicates the how many EPR pairs the subsystem $A \cup B$ loses in the non-equilibrium process. The initial value of $I_{A,B,\overline{B}}$ will be zero because the initial value of BOMI is determined by the number of EPR pairs. In other words, since no EPR pairs are delocalized, the value of $I_{A,B,\overline{B}}$ is zero. The negative value of $I_{A,B,\overline{B}}$ suggests how many number of EPR pairs are delocalized and locally hidden by the dynamics. For $|\sigma_\alpha^{(i)}(t)\rangle$, at $t = 0$, the entanglement structure of a single EPR on $i$-th site is deformed by $\sigma_\alpha^{(i)}(t = 0)$, and we expect the time evolution of $I_{A,B}$ and $I_{A,B,\overline{B}}$ to be interpreted in the same manner.

- **Partial overlap configuration (Fig. 2 (b))** As explained above, in this configuration of 2D CFTs, the time evolution of $I_{A,B}$ and $I_{A,B,\overline{B}}$ strongly depends on the scrambling ability. The 2D holographic CFT is the theory with the large degrees of freedom and strong scrambling effect. Although the spin systems which we consider in this paper does not have the large degrees of freedom, we expect the time evolution of $I_{A,B}$ and $I_{A,B,\overline{B}}$ in this configuration to strongly depend on the scrambling ability of the dynamics. Therefore, we will consider the partial overlap configuration.

- **Disjoint configuration (Fig. 2 (c))** In this configuration, there are no EPR pairs, both edges of which are in $A \cup B$. Then, the initial value of $I_{A,B}$ is zero. For $|U(t)\rangle$ and $\sigma_\alpha^{(i)}(t)$, OEEs for $A$ and $B$ are independent of time. The time evolution of $I_{A,B}$ is determined by that of $S_{A \cup B}$. The dynamics with strong scrambling makes the entanglement structure of $A \cup B$ less structure-less, so that the value of $S_{A \cup B}$ is expected to increases in the non-equilibrium process. Since the initial value of $S_{A \cup B}$ is the one for the maximally entangled state, we expect that the value of $I_{A,B}$ does not vary in time so much.

# 3 BOMI and TOMI in a spin chain

We study the dynamics of the spin chain,

$$H = \sum_{i=1}^{L} \left[ \sigma_z^{(i)} \sigma_z^{(i+1)} + h_x \sigma_x^{(i)} + h_z \sigma_z^{(i)} \right], \tag{27}$$

where $L$ is the number of site of this system, and $\sigma_a^{(i)}$ are the Pauli spin operators on site $i$. For periodic boundary conditions, we define $\sigma_a^{(L+1)} = \sigma_a^{(1)}$, whereas for open boundary conditions, the term involving $\sigma_a^{(L+1)}$ is ignored. The dynamics of the Hamiltonian with $(h_x, h_z) = (-1.05, 0.5)$ is chaotic [33, 37].

The authors in [14] studied the time evolution of the BOMI and TOMI for the unitary operator $U(t)$, and in particular, studied the scrambling ability of the chaotic chain. In this paper, we study the scrambling ability of the chain in terms of local operators, and compare the BOMI and TOMI of local operators to $U(t)$. We consider the time evolution for Pauli's operators in the Heisenberg picture $\sigma_a^{(i)}(t) = e^{iHt} \sigma_a^{(i)} e^{-iHt}$ for a spin chain with periodic and open boundary conditions.

We investigate the time evolution of the BOMI and TOMI in the integrable and chaotic phases. We briefly summarize the results, and we will give some interpretation on these results later:

1. **Integrable phase:** For integrable cases, we study both longitudinal, $(h_x, h_z) = (0, 2)$, and transverse, $(h_x, h_z) = (0.5, 0)$, magnetic fields [67]. We focus on the longitudinal case in the main text. We consider the transverse case in Appendix. A. In the longitudinal case, $H$ and $\sigma_z^{(i)}$ commute. Therefore, the BOMI and TOMI of $\sigma_z^{(i)}$ are independent of time. The BOMI and TOMI of $\sigma_x$ and $\sigma_y$ are either constant or periodic functions in time, depending on the location of the operator and the boundary conditions. The BOMI and TOMI of $U(t)$ is always a periodic function in time, but the amplitude depends on the boundary conditions.

2. **Chaotic phase:** For the parameters $(h_x, h_z) = (-1.05, 0.5)$, the system is chaotic.

   - *Early-time evolution:* An effective light cone picture explained in section 3.1 qualitatively describes the early-time evolution of the BOMI and TOMI of the local operators for the full and partial overlap configurations. The value of BOMI in the disjoint configuration is very small, which implies that little information spreads.
   - *Late-time evolution:* The late-time values for the full and partial overlap configurations are independent of the operators and boundary conditions and become the same as the unitary operator. The Page curve, explained in 3.2, describes the late-time values of the BOMI for these configurations. The value of the BOMI in the disjoint configuration is very small since it was initially small.

Although we studied the time evolution of TOMI and BOMI for the full overlap, partial overlap, and disjoint configurations, we found that even in the full overlap configuration, the time evolution of BOMI and TOMI can capture the dynamical properties of the integrable and chaotic phases. Therefore, we will explain only the full overlap case in the main text, and the others will be explain in Appendix. A. In the following subsections, we explain the time evolution of the BOMI and TOMI in detail, and will give the interpretation on these behaviors in time.

## 3.1 Early-time evolution

For the three configurations in Figure 2, we divide the input (original) system of size $L$ into $A$ and $\bar{A}$ of size $l_A$ and $L - l_A$, and the output (scattered) system, also of size $L$, into $B$ and $\bar{B}$

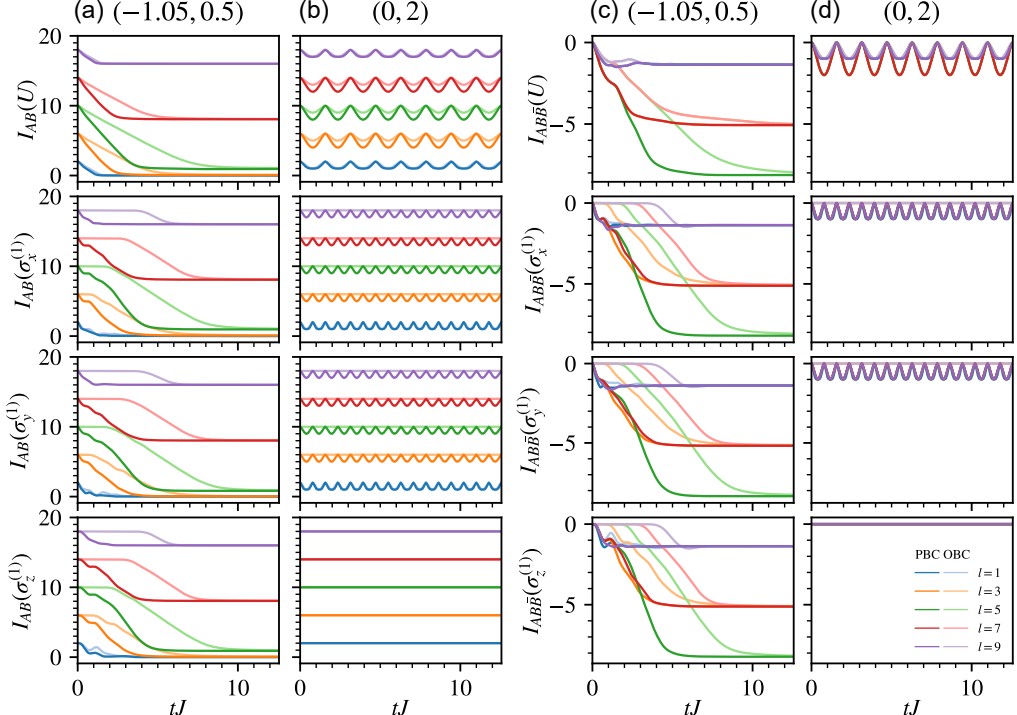

Figure 3: Time evolution of the (a,b) BOMI and (c,d) TOMI of the operators $U$, $\sigma_x^{(1)}$, $\sigma_y^{(1)}$, and $\sigma_z^{(1)}$ in the chaotic chain model for the fully overlapping configuration. The local operators are located at the left boundary of $A$ and $B$, and the sizes of $A$ and $B$ are equal, $l_A = l_B = l$. The magnetic field is $(h_x, h_z) = (-1.05, 0.5)$ for (a,c) and $(h_x, h_z) = (0, 2)$ for (b,d). Dark lines correspond to periodic boundary conditions and light lines correspond to open boundary conditions. The total number of sites is $L = 10$.

of size $l_B$ and $L - l_B$. We consider periodic boundary conditions (PBC) and open boundary conditions (OBC).

In Figure 3, we show the time evolution of the BOMI in panels (a,b) and TOMI in panels (c,d) for the full overlap configuration, as shown in Figure 2(a). The BOMI and TOMI are defined by (25) and (26). In this configuration, $l_A = l_B = l$. The local operators $\sigma_{a=x,y,z}^{(1)}$ are at the leftmost site of $A$ and $B$. In the open spin chain, the local operator site is also the leftmost site of the system. In panels (a,b), the BOMI always starts at $2l$. This is equal to the number of EPR pairs contained in $A \cup B$. In the chaotic phase in panel (a), the BOMI begins to decrease instantly, but the slope depends on the boundary conditions. This is explained by a light cone picture. The BOMI then settles on a stable value, described by the Page curve. Both the light cone and the Page curve will be described in more detail in the next sections. The BOMI for the integrable phases is shown in panels (b). For longitudinal field, shown in (c), the BOMI for $\sigma_z$ is always constant and the BOMI for $U$ oscillates. For $\sigma_x$ and $\sigma_y$, the BOMI oscillates for PBC and is constant for OBC except for $l = 1$ where it is constant.

The TOMI always starts at zero. In the chaotic phase, shown in (c), the TOMI decreases and settles on a value depending on $l$. The late-time value is the same for $l$ and $L - l$. For the longitudinal field in (f), the TOMI shows consistent oscillatory behavior below zero or stays constant at zero.

As in [68], quantum revival occurs in a two-dimensional free field theory on a compact manifold. Here, quantum revival refers to the BOMI returning to its initial value. The revival time is proportional to the system size, because the revival in the free field theory follows the

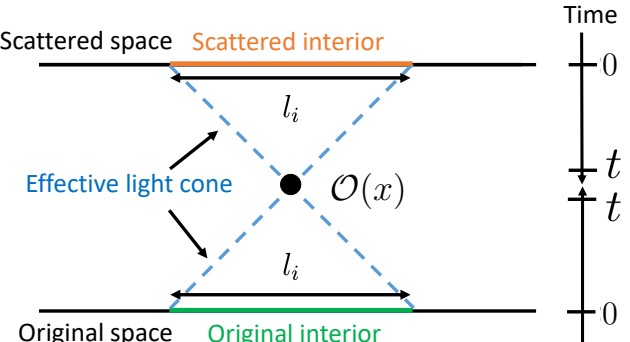

Figure 4: Schematic of the effective light cone for a local operator $\mathcal{O}(x, t)$. The interior of the light cone grows over time. Information in the interior of the light cone of the original space is communicated through the local operator to the interior of the light cone of the scattered space. The local operator is approximately identity in the exterior of the light cone.

relativistic propagation of quasi-particles. The BOMI for PBC in the chaotic phase does not return to the original one. Therefore, a revival does not occur for PBC in the chaotic phase.

**Effective light cone picture**

We find the time evolution of the BOMI and TOMI of local operators in the chaotic phase qualitatively follows an effective light cone picture. Let us explain the effective light cone picture, which well describes the distinctive behaviors of the early-time BOMI and TOMI of Pauli's spin operators in the chaotic phase of (27). Consider a local operator in the Heisenberg picture $\mathcal{O}(x, t)$. As in Figure 4, we call the original (scattered) subspace in the light cone *original (scattered) interior*. The interior of effective light cone is the region where $\mathcal{O}(x, t)$ has influence: The local operator makes changes to the entanglement structure of the scattered subspace. The exterior of the light cone is approximately equal to a local identity. The size of the original and scattered subsystems inside of the light cone, $l_i$, depends on an effective velocity, $v_{\text{eff}}$, determined by the details of the theory. When $v_{\text{eff}}$ is a constant velocity, $l_i$ is determined by $v_{\text{eff}}$,

$$l_i = 2v_{\text{eff}}t. \tag{28}$$

Here, we divide the original Hilbert space into $A$ and $\bar{A}$, while we divide the scattered Hilbert space into $B$ and $\bar{B}$. In this picture, if $A_l$ and $B_l$ ($A_r$ and $B_r$) are in the light cone and $A_r$ and $B_r$($A_l$ and $B_l$) are not as in (a) of Figure 5((b) of Figure 5), then BOMI and TOMI decreases. In the early time interval, as in panels (a) and (b) of Figure 5, $A_l$ and $B_l$ ($A_r$ and $B_r$) are in the light cone and $A_r$ and $B_r$($A_l$ and $B_l$) are not, so that $I_{A,B}$ decreases. After enough time, as in the panes (c) and (d) of Figure 5, $A_l$, $B_l$, $A_r$, and $A_l$ are in the light cone, so that the slope of $I_{A,B}$ flattens. In Figure 6, we show the light cone picture effectively describes the time evolution of $I_{A,B}$. For the disjoint configuration, $A$ is almost uncorrelated to $B$, initially. Then, $I_{A,B}$ becomes small compared to the other configurations, as seen in Figure 12. In two-dimensional holographic CFTs, which are theories that have a gravity dual, the time evolution of operator BOMI and TOMI is well-described by a very simple explanation, the line tension picture [23].

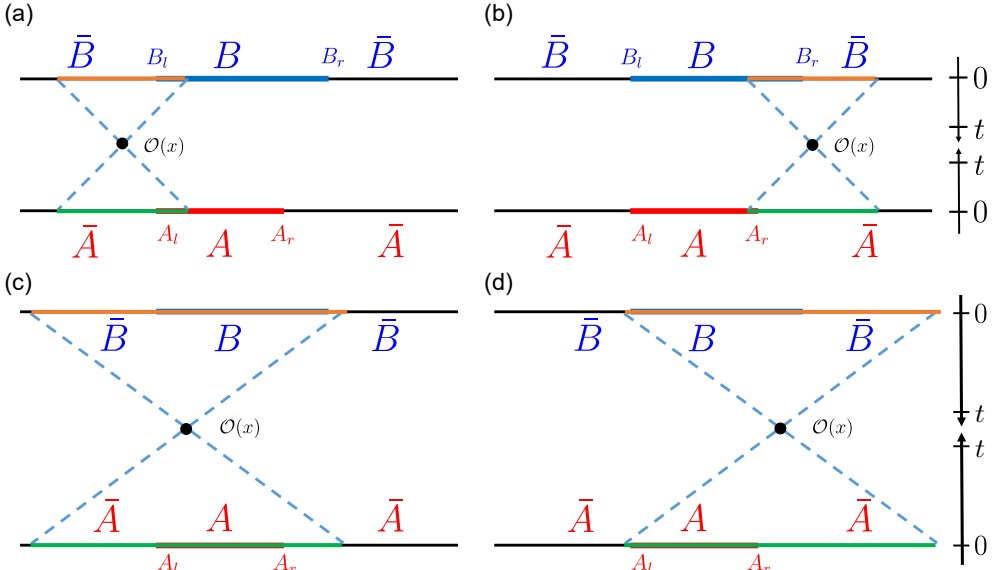

Figure 5: Schematic of the effective light cone in the partially overlapping configuration.

## 3.2 Late-time evolution

Here, we will explain the most important results of the spin system (27) in the chaotic phase.

**Chaotic chain**

Consider the chaotic phase of (27). The late-time evolution of the BOMI and TOMI of the local operators is independent of operators and boundary conditions, and is approximately equal to that of the BOMI and TOMI of the unitary operator. This suggests that while the entanglement structure of the state may depend on the local operators and boundary conditions for the intermediate time region, the entanglement structure of the late time state may not. In other words, the strong scrambling effect of the dynamics washes out the information about the local operators and boundary conditions, so that $|\sigma_{\alpha^{(i)}=x,y,z}\rangle$ with the various boundary conditions flows to the typical state, the state which is independent of the entanglement structure of the initial state. To confirm this suggestion, we will compare the late time behavior of BOMI and TOMI in the chaotic chain to that following the Page curve. The entanglement entropy for a typical state is expected to follow the Page curve. In this case, the Page curve is a function of the sizes of input and output (original and scattered) subsystems.

**The Page curve**

The Page curve in this paper will be the effective description of OEEs for a typical state. The OEE of the local operator state in (21) for the original subsystem $A$ (scattered subsystem $B$) is proportional to the subsystem size. It is expected that if the dynamics has strong scrambling ability, then the late-time state is structureless. Therefore, the late-time OEE for a subsystem $\mathcal{V}$ with total size $\hat{L}$ is expected to be the entanglement entropy for the maximally entangled state:

$$S_{\mathcal{V}} = \begin{cases} \hat{L}, & 0 \leq \hat{L} < L, \\ L - \alpha, & \hat{L} = L, \\ 2L - \hat{L}, & L < \hat{L} \leq 2L, \end{cases} \tag{29}$$

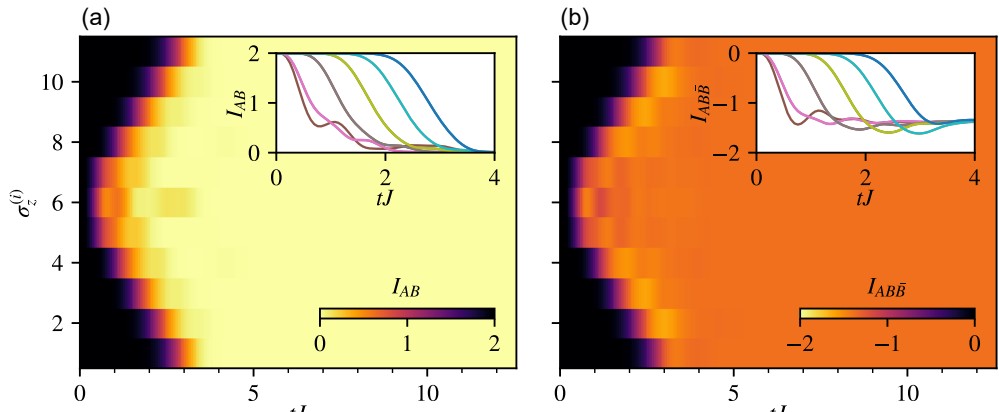

Figure 6: Time evolution of (a) BOMI and (b) TOMI for the local operator $\sigma_z^{(i)}$ at different sites in the chaotic chain model for the fully overlapping configuration with open boundary conditions. The y-axis corresponds to the location of the operator. Inset shows the same information with (a) BOMI and (b) TOMI on the y-axis. The subsystems, $A$ and $B$, are centered at site 6 with size $l_A = l_B = 1$ and total size $L = 11$. The magnetic field is $(h_x, h_z) = (-1.05, 0.5)$.

where $\hat{L}$ is an even integer. The parameter $\alpha$ depends on the parameters $h_x$ and $h_z$. If we apply (29) to (25), we get the late time value:

$$I_{A,B} = \begin{cases} 0, & 0 \le \hat{L} < L, \\ \alpha, & \hat{L} = L, \\ 2(\hat{L} - L), & L < \hat{L} \le 2L. \end{cases} \tag{30}$$

Since $I_{A,B}$ has to be non-negative, $\alpha$ must also be a non-negative number.

We plot the late-time value of $S_{A,B}$ for the full and partial overlap configurations in Figure 7, where $s = 0$ corresponds to the full overlap configuration. The OEEs for $A$ and $B$ is given by $l_A = l$ and $l_B = l + s$, respectively. Then, the behavior of $I_{A,B}$ following the plot of Figure 7 is consistent with (30) with $\hat{L} = 2l + s$. For the full overlap configuration, $\hat{L} = 2l$. Thus, when $L$ is odd, we are not able to take $2l$ to be $L$. For the disjoint configuration, the size of $A \cup B$ is less than or equal to $L$ since $A$ and $B$ are not overlapping. Therefore, $I_{A,B}$ should vanish at late time according to (29). However, in the numerical computation, $I_{A,B}$ does not vanish, but is a small value, $\beta$. Thus, $\beta$ is not described by the Page curve.

Applying (30) to (26), we get the late-time value of $I_{A,B,\bar{B}}$ in Figure 2(a):

$$I_{A,B,\bar{B}} = \begin{cases} \alpha - 2l, & 0 \le 2l < L, \\ 2\alpha - 2l, & 2l = L, \\ \alpha + 2(l - L), & L < 2l \le 2L, \end{cases} \tag{31}$$

where $L$ is an even integer, for simplicity.

**Thermodynamic limit**

Let us consider the late-time value of BOMI in the full overlap, partial overlap, and disjoint configurations, and TOMI in thermodynamic limit where we take $L$, the system size, to be infinite. By using (29) in the Page curve picture, the late-time value of BOMI in the full overlap, partial overlap, and disjoint configurations is equal to zero. Then, the late-time value of first

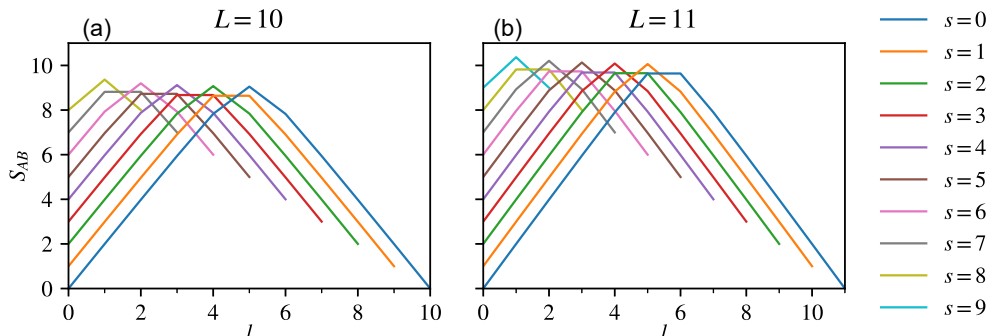

Figure 7: Late time entanglement entropy of the operator $U(t)$ in the chaotic chain model for the partially overlapping configuration (fully overlapping when $s = 0$). We show the entanglement entropy at time $t = 10^{12}$ for an (a) even system size $L = 10$ and (b) odd system size $L = 11$. The size of $A$ is $l_A = l$ and $B$ is $l_B = l + s$. The magnetic field is $(h_x, h_z) = (-1.05, 0.5)$. We show the entanglement entropy for periodic boundary conditions and operator $U(t)$ but note that the late time entanglement entropy is the same for open boundary conditions and operators $\sigma_x^{(i)}$, $\sigma_y^{(i)}$, and $\sigma_z^{(i)}$.

and second terms of (26) is zero, whereas that of the last term is constant, so that the late-time value of TOMI is

$$I_{A,B,\bar{B}} \to -2l \,. \tag{32}$$

The late-time value of TOMI, $-2l$, is consistent with that of the TOMI of a unitary operator in two-dimensional holographic CFT [21–23]. The late time values in (31) and (32) show that finite size effects prevent the dynamics from delocalizing quantum information since the late time value of TOMI in an infinite space is smaller in magnitude compared to a finite space.

### 3.2.1 Factorization of the density matrix

Finally, we will interpret the late time behavior of BOMI in terms of the reduced density matrix. For simplicity, we consider the late-time behavior of $I_{A,B}$ in the full overlap configuration. We assume $l_A + l_B < L$. In the chaotic phase, regardless of the states and boundary conditions, the late time value of $I_{A,B}$ is zero. This suggests that $\rho_{A \cup B}$ in the late time region can be approximately factored into $\rho_A$ and $\rho_B$:

$$\rho_{A \cup B} \underset{t \gg 1}{\approx} \rho_A \otimes \rho_B \,. \tag{33}$$

Thus, the correlation between $A$ and $B$ is lost or weak in the time evolution induced by the chaotic spin chain.

## 4 Time evolution of BOMI and TOMI in a disordered spin chain

In this section, we study the BOMI and TOMI of $U(t)$, $\sigma_x^{(i)}$, $\sigma_y^{(i)}$, and $\sigma_z^{(i)}$ in a spin system with disorder:

$$H_{\mathrm{MBL}} = \sum_{i=1}^{L} J \vec{\sigma}^{(i)} \cdot \vec{\sigma}^{(i+1)} + \sum_{i=1}^{L} h_i \sigma_z^{(i)} \,, \tag{34}$$

where $\vec{\sigma}^{(i)}$ is the vector of Pauli spin operators and $h_i$ is a disorder parameter drawn from a uniform random distribution, $h_i \in [-w, w]$. Hereafter we set $J = 1$ as the unit of energy. The

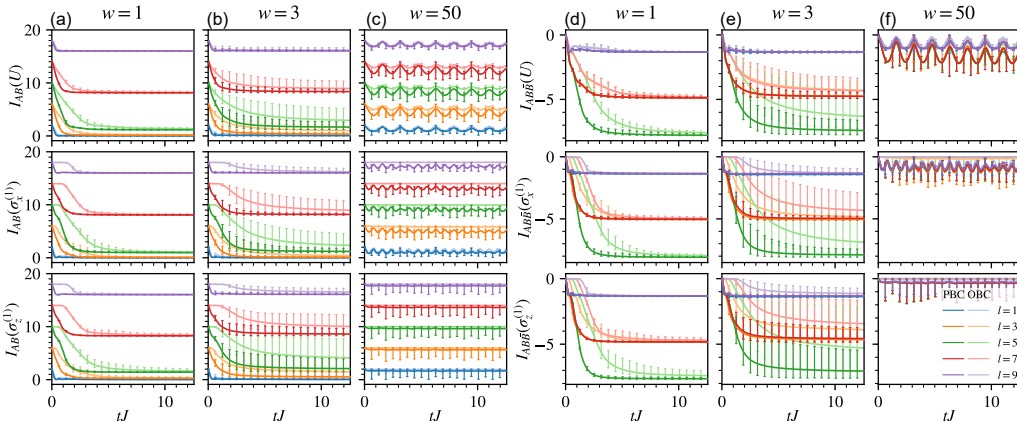

Figure 8: Time evolution of (a-c) BOMI and (d-f) TOMI of the operators $U(t)$, $\sigma_x^{(1)}$, $\sigma_y^{(1)}$, and $\sigma_z^{(1)}$ in the disordered spin chain model for the fully overlapping configuration. The local operators are located at the left boundary of $A$ and $B$ and the size of $A$ and $B$ are equal, $l_A = l_B = l$. The disorder strength is (a,d) $w = 1$, (b,e) $w = 3$, and (c,f) $w = 50$. Solid lines show the mean over 20 disorder configurations and error bars show the min and max. Dark lines correspond to periodic boundary conditions and light lines correspond to open boundary conditions. The total number of sites is $L = 10$.

dynamical features of the disordered spin chain are mostly captured by the BOMI and TOMI in the full overlap configuration, which we will explain in the following section. We consider the partial overlap and disjoint configurations in Appendix B.

The time evolution of BOMI and TOMI for small, near critical, and large $w$, are show in Figure 8.

- **Small $w$ region ($w = 1$):** As in [41, 69–72], (34) with small $w \ll 4$ is in the chaotic phase. Therefore, the late-time value of BOMI and TOMI is independent of operators and boundary conditions, and is well-described in terms of the Page curve (See, Figure 8.).

- **Near critical region ($w = 3$):** For $w = 3$, near the critical point,[1] the time evolution of BOMI and TOMI depends on the operators and boundary conditions. In the full overlap configuration, the late time value of BOMI and TOMI is larger than that in (34) with $w = 1$. When $l_A + l_B < L$, the late behavior of BOMI suggests the dynamics of the disordered spin chain with $w = 3$ prevents $\rho_{A \cup B}$ from factorizing to $\rho_A$ and $\rho_B$.

- **Large $w$ region ($w = 50$):** In the large $w$ region ($w = 50$), BOMI and TOMI of $\sigma_z(t)$ are almost time-independent. This is because $\sigma_z^{(i)}(t)$ is well approximated by

$$\sigma_z^{(i)}(t) \approx e^{iH_{\mathrm{eff}}t} \sigma_z^{(i)} e^{-iH_{\mathrm{eff}}t} = \sigma_z^{(i)}, \tag{35}$$

where $H_{\mathrm{eff}}$ is described in the next section. For other operators, we see that the BOMI and TOMI have oscillating behavior. We explain these oscillations using an effective Hamiltonian in the next section. Moreover, BOMI and TOMI increases with $w$ for all operators in the large $w$ region.

---

[1]Recent progress [60] has shown that in the thermodynamic limit, the system may be in the MBL phase if the strength of the disorder is $w > 18$. However, for the size of the system studied in this paper, around $w = 3$, the behavior of the physical quantities changes from chaotic to that in MBL. Therefore, we will study the time evolution of BOMI and TOMI at $w = 3$.

## 4.1 Effective theory in the strong disordered phase.

Consider the limit $J \ll w$ with OBC for simplicity. The energies of (34) to first order in $J/w$ are

$$E \approx J \sum_{i=1}^{L-1} \sigma_z^{(i)} \sigma_z^{(i+1)} + \sum_{i=1}^{L} h_i \sigma_z^{(i)}, \tag{36}$$

which are the same energies as the disordered Ising chain. We consider the effective Hamiltonian

$$H_{\text{eff}} = J \sum_{i=1}^{L-1} \sigma_z^{(i)} \sigma_z^{(i+1)} + \sum_{i=1}^{L} h_i \sigma_z^{(i)}. \tag{37}$$

The effective Hamiltonian is diagonal in the $\sigma_z^{(i)}$ basis and the entanglement entropy is exactly solvable. To demonstrate this, consider the local operator $\sigma_x^{(1)}$. The operator in the eigenbasis of $H_{\text{eff}}$ is given by

$$\sigma_x^{(1)} = \sum_{\{\sigma_z, \tau_z\}} |\sigma_z^{(1)} \dots \sigma_z^{(L)}\rangle e^{i(H_{\text{eff}}[\{\sigma_z\}] - H_{\text{eff}}[\{\tau_z\}])t} \langle \sigma_z^{(1)} \dots \sigma_z^{(L)} | \sigma_x^{(1)} | \tau_z^{(1)} \dots \tau_z^{(L)} \rangle \langle \tau_z^{(1)} \dots \tau_z^{(L)} |$$

$$= \sum_{\{\sigma_z\}} |\sigma_z^{(1)} \dots \sigma_z^{(L)}\rangle \exp\left\{2it\left[J\sigma_z^{(1)}\sigma_z^{(2)} + h_1\sigma_z^{(1)}\right]\right\} \langle \bar{\sigma}_z^{(1)} \dots \sigma_z^{(L)} |,$$

where the bar denotes flipping the spin. The density matrix for the dual state to $\sigma_x^{(1)}$ is given by

$$\rho = \frac{1}{2^L} \sum_{\{\sigma_z, \tau_z\}} \exp\left\{2it\left[J\left(\sigma_z^{(1)}\sigma_z^{(2)} - \tau_z^{(1)}\tau_z^{(2)}\right) + h_1\left(\sigma_z^{(1)} - \tau_z^{(1)}\right)\right]\right\}$$

$$\times |\sigma_z^{(1)} \dots \sigma_z^{(L)}\rangle \langle \tau_z^{(1)} \dots \tau_z^{(L)}|_{\text{in}} \otimes |\bar{\sigma}_z^{(1)} \dots \sigma_z^{(L)}\rangle \langle \bar{\tau}_z^{(1)} \dots \tau_z^{(L)}|_{\text{out}}.$$

We see that if both $\sigma_z^{(1)}$ and $\sigma_z^{(2)}$ are traced out in either the input or output space, then the exponential becomes unity and the density matrix becomes constant in time. If we trace out either $\sigma_z^{(1)}$ or $\sigma_z^{(2)}$, then we get a cosine dependence on time. As an example, we show the mutual information when $A$ and $B$ are the original and scattered spaces that only includes site 1, $A = \mathcal{H}_{1_{\text{in}}}$ and $B = \mathcal{H}_{1_{\text{out}}}$. For subsystems that are not overlapping in the input and output spaces, we trace out both $\sigma_z^{(1)}$ and $\sigma_z^{(2)}$ and the exponential becomes unity. Thus, the reduced density matrix and entanglement entropy is given by

$$\rho_A = \frac{1}{2}\mathbb{1}_{\mathcal{H}_A}, \qquad \rho_B = \frac{1}{2}\mathbb{1}_{\mathcal{H}_B}, \qquad \rho_{\bar{A}} = \frac{1}{2^{L-1}}\mathbb{1}_{\mathcal{H}_{\bar{A}}},$$

$$\rho_{\bar{B}} = \frac{1}{2^{L-1}}\mathbb{1}_{\mathcal{H}_{\bar{B}}}, \qquad \rho_{A \cup \bar{B}} = \frac{1}{2^L}\mathbb{1}_{\mathcal{H}_A} \otimes \mathbb{1}_{\mathcal{H}_{\bar{B}}}, \qquad \rho_{B \cup \bar{B}} = \frac{1}{2^L}\mathbb{1}_{\mathcal{H}_B} \otimes \mathbb{1}_{\mathcal{H}_{\bar{B}}},$$

$$S_A = 1, \qquad S_B = 1, \qquad S_{\bar{A}} = (L-1),$$

$$S_{\bar{B}} = (L-1), \qquad S_{A \cup \bar{B}} = L, \qquad S_{B \cup \bar{B}} = L.$$

For the overlap, the reduced density matrix and entanglement entropy are given by

$$\rho_{A \cup B} = \frac{1}{2} \sum_{\sigma_z^{(1)}, \tau_z^{(1)}} \cos\left(2tJ\left(\sigma_z^{(1)} - \tau_z^{(1)}\right)\right) \exp\left[2ith_1\left(\sigma_z^{(1)} - \tau_z^{(1)}\right)\right] |\sigma_z^{(1)}\rangle \langle \tau_z^{(1)}|_{\text{in}} \otimes |\bar{\sigma}_z^{(1)}\rangle \langle \bar{\tau}_z^{(1)}|_{\text{out}},$$

$$S_{A \cup B} = -\cos^2(2tJ)\log_2(\cos^2(2tJ)) - \sin^2(2tJ)\log_2(\sin^2(2tJ)).$$

Therefore,

$$I_{A,B} = 2 - S_{A \cup B}, \qquad I_{A,\bar{B}} = 0, \qquad I_{A,B,\bar{B}} = -S_{A \cup B}.$$

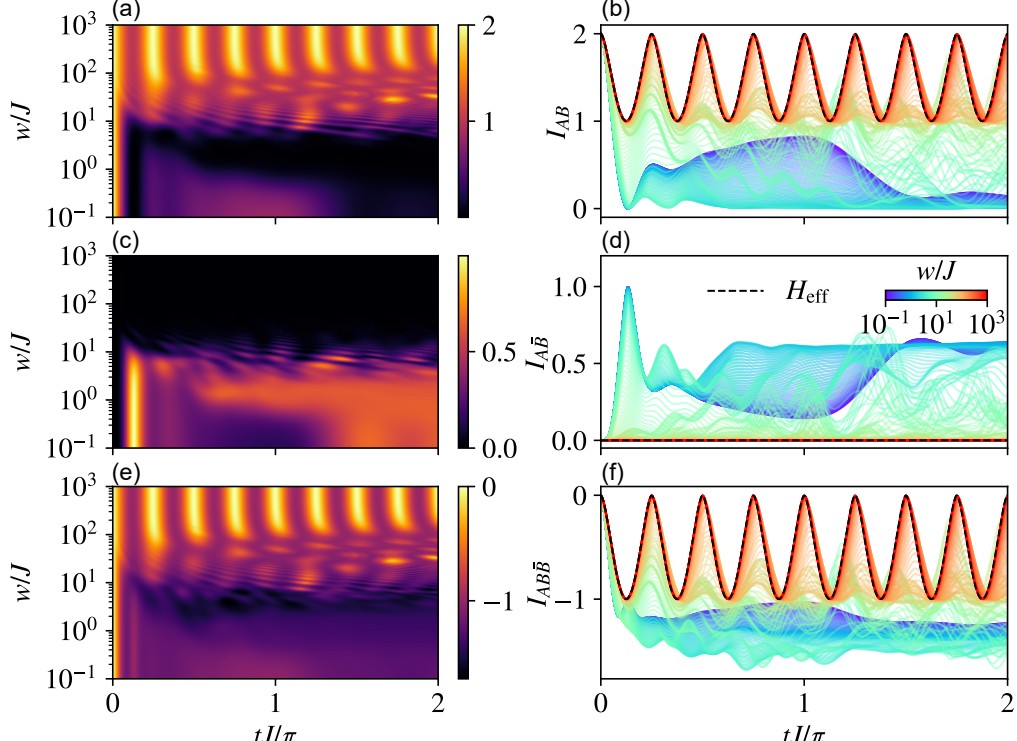

Figure 9: Time evolution vs. disorder strength of (a-d) BOMI and (e-f) TOMI of the operator $\sigma_x^{(1)}$ in the disordered spin chain model for the fully overlapping configuration. We use a single disorder configuration to show the evolution over varying disorder strengths. Dashed lines in (b,d,f) show the BOMI and TOMI using the effective Hamiltonian, corresponding to the $J \ll w$ regime of the disordered spin chain. The subsystems, $A$ and $B$, are centered at site 1 with size $l_A = l_B = 1$ and total size $L = 10$ using open boundary conditions.

In Figure 9, we show the time evolution of BOMI and TOMI for various disorder strengths. The BOMI and TOMI are well described by the effective Hamiltonian for $w/J > 10^2$. For strong disorder, the oscillation of BOMI and TOMI depends only on the local interaction $J$, even though the reduced density matrix depends on not only $J$, but also the disorder, $h_i$. We can follow a similar procedure to get the density matrix for different operators.

$$\rho\left(\sigma_z^{(i)}\right) = \frac{1}{2^L} \sum_{\{\sigma_z, \tau_z\}} \sigma_z^{(i)} |\sigma_z^{(1)} \dots \sigma_z^{(L)}\rangle \langle \tau_z^{(1)} \dots \tau_z^{(L)}|_{\text{in}} \otimes |\sigma_z^{(1)} \dots \sigma_z^{(L)}\rangle \langle \tau_z^{(1)} \dots \tau_z^{(L)}|_{\text{out}} ,$$

$$\rho(U) = \frac{1}{2^L} \sum_{\{\sigma_z, \tau_z\}} e^{-i(H_{\text{eff}}[\{\sigma_z\}] - H_{\text{eff}}[\{\tau_z\}])t} |\sigma_z^{(1)} \dots \sigma_z^{(L)}\rangle \langle \tau_z^{(1)} \dots \tau_z^{(L)}|_{\text{in}}$$

$$\otimes |\sigma_z^{(1)} \dots \sigma_z^{(L)}\rangle \langle \tau_z^{(1)} \dots \tau_z^{(L)}|_{\text{out}} .$$

$\sigma_x^{(1)}$ and $\sigma_y^{(1)}$ are related by a rotational symmetry of the spins around the $\hat{z}$ axis. The density matrix for the $\sigma_z^{(i)}$ operators do not have any time dependence and any BOMI or TOMI will be constant. For the time-evolution operator, we see that if all the $\sigma_z$ and $\tau_z$ are the same, then the BOMI and TOMI will be constant. Therefore, we only see oscillations for the fully overlapping and partially overlapping cases.

Thus, the periodic behavior in time of BOMI and TOMI with large $w$ is due to the off-diagonal part of the density matrix expanded by the eigenstates of $\sigma_z^{(i)}$. This periodic behavior suggests the state may return back to the initial one and the quantum nature of the state

may be preserved by the time evolution induced by the disordered spin chain with strong disorder. The time evolution of BOMI and TOMI can be periodic in time with the period $\frac{\pi}{4J}$. Thus, the BOMI and TOMI exhibits quantum revival. In 2D non-holographic CFT on compact space, the time evolution of BOMI will exhibit the quantum revival followed by the relativistic propagation of quasi-particles. The period of the quantum revival is proportional to the system size [7, 32, 68, 73, 74]. However, in this spin chain, the period is determined by the strength of local interaction. Therefore, even if we take $L$ to be infinite, the BOMI and TOMI may periodically evolve in time. The value of BOMI in Figure 9 is between one and two. Here, the size of the subsystems, $A$ and $B$, is one. In the EPR picture, the value of BOMI will be two or zero. Therefore, we can not explain the time evolution of BOMI in terms of EPR pairs. This suggests that something beyond the quasi-particle picture is needed to explain the entanglement dynamics in the MBL phase.

## 5 Discussions and future directions

### Discussions

Let us summarize the results, compare with the CFT results [21–23], and comment on a few future directions.

### Summary

We have studied the time evolution of BOMI and TOMI for the unitary time evolution operator and Pauli's spin operators in the chaotic chain (27) and MBL chain (34).
  We found that:

- **Chaotic chain**: The early-time evolution of TOMI and BOMI is qualitatively described in terms of an effective light cone picture. Their late-time evolution is independent of the boundary condition and operators, and their late-time value is quantitatively explained by the Page curve. For PBC, quantum revival does not occur and the compactness of space time prevents the dynamics from delocalizing the quantum information.

- **Disordered spin chain**: In the weak disordered system ($w = 1$), the late-time value of BOMI and TOMI is well-described by the Page curve. When the disorder is large, BOMI and TOMI in full overlap configuration increases when $w$ increases. In the very strong disorder system ($w = 50$), BOMI and TOMI in some of configurations oscillate with respect to time. This oscillation is well-described by an effective Hamiltonian.

### Comparison with CFTs

Here, we compare the late time value of BOMI and TOMI derived by the Page curve with that of holographic CFTs [21–23], which are theories that have a gravity dual. After taking the thermodynamic limit, the Page curve describes the late time value of BOMI. As in (32), when we take the thermodynamic limit, the late-time value of TOMI in the chaotic phase of (27) is

$$I(A, B, \bar{B}) = -2l \,, \tag{38}$$

where 2 counts the dimension of the local Hilbert space. In the holographic CFT, the late-time value of TOMI is

$$I_{A,B,\bar{B}} = -2l \log_2 \left[ e^{\left( \frac{\pi c}{6\epsilon} \right)} \right], \tag{39}$$

where $c$ is the central charge, and $\epsilon$ is the scale which characterizes the initial state, $|\Psi\rangle_{\text{initial}} \propto \sum_a^{\frac{1}{\epsilon}} C_a e^{-itE_a} |a\rangle$. Therefore, we expect the chaotic chain with approximately $e^{\left(\frac{\pi c}{6\epsilon}\right)}$ local degrees of freedom to mimic the dynamical properties of holographic CFT.

**Future directions**

Let us comment on a few future directions:

1. An important future direction is to construct a tractable system with a large number of local degrees of freedom and strong scrambling ability in order to mimic holographic CFTs.

2. To compute logarithmic negativity and reflected entropy is also interesting. In holographic CFTs, these values show phase transition-like behavior [20–23]. It would be interesting to see whether logarithmic negativity or reflected entropy in the chaotic chain also show such a behavior.

3. We numerically found that the late time value of TOMI in the chaotic chain on the compact manifold was non-vanishing. It would be interesting to analytically compute this in a two-dimensional holographic CFT on a compact manifold to find the finite size correction compared to a non-compact holographic CFT.

# Acknowledgments

We would like to thank Shinsei Ryu, Jonah Kundler-Flam, and Mao Tian Tan for useful discussions. M.N. and M.T. thank the Yukawa Institute for Theoretical Physics at Kyoto University for hospitality during the workshops YITP-T-18-04 "New Frontiers in String Theory 2018" and YITP-T-19-03 "Quantum Information and String Theory 2019".

**Funding information** The work of M.T. was partially supported by Grant-in-Aid No. JP17K17822, No. JP20H05270, No. JP20K03787, and JP21H05185 from JSPS of Japan. The work of M.N. was partially supported by Grant-in-Aid No. JP19K14724 from JSPS of Japan.

# A The time evolution of TOMI and BOMI in the one-dimensional Ising model with magnetic field

Here, we will explain the time evolution of BOMI and TOMI in the partial overlap and disjoint configurations of the one-dimensional Ising model with magnetic field.

In Figure 10, we show the time evolution of TOMI and BOMI in the full overlap configuration in the one-dimensional Ising model $(0.5, 0)$. The parameters $(h_x, h_z) = (0.5, 0)$ are also in the integrable phase. However, the BOMI and TOMI of all operators are neither constant nor periodic functions. Even the late-time evolution of the BOMI and TOMI depends on the operators and boundary conditions. For transverse field, shown in (b), the dynamics has some scrambling ability and shows characteristics similar to (a), but also has some oscillatory behavior similar to (c). For transverse field in (e), the TOMI shows unpredictable behavior, but is in general, more negative for PBC.

In Figure 11, we show the time evolution of the BOMI and TOMI for the partial overlap configuration. In this configuration, the number of sites in $B$ is larger than $A$ by $s$. The size of $B$ is equal to $l_B = l_A + s$ In Figure 11, we show the time evolution of $I_{A,B}$ and $I_{A,B,\bar{B}}$ for

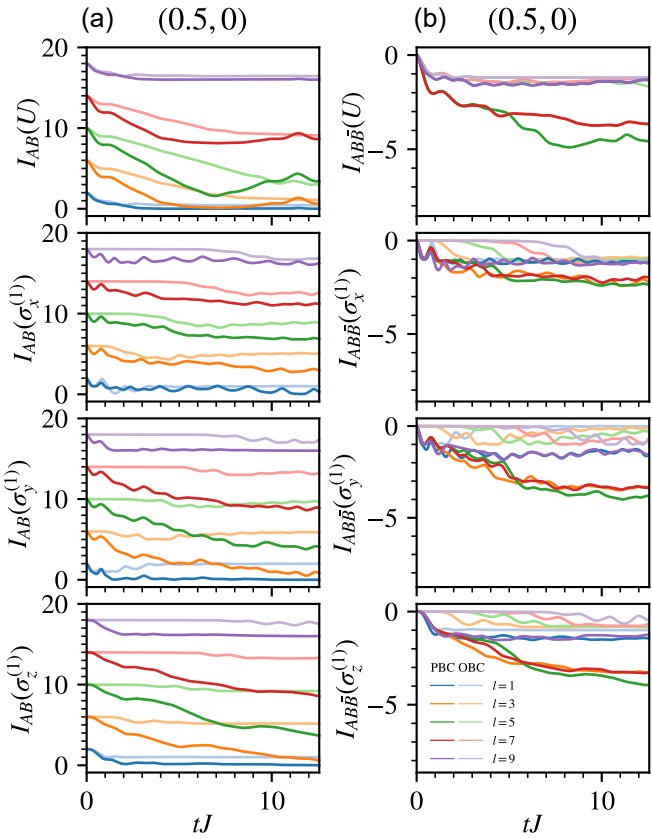

Figure 10: Time evolution of the (a) BOMI and (b) TOMI of the operators $U$, $\sigma_x^{(1)}$, $\sigma_y^{(1)}$, and $\sigma_z^{(1)}$ in the chaotic chain model for the fully overlapping configuration. The local operators are located at the left boundary of $A$ and $B$, and the sizes of $A$ and $B$ are equal, $l_A = l_B = l$. The magnetic field is $(h_x, h_z) = (0.5, 0)$. Dark lines correspond to periodic boundary conditions and light lines correspond to open boundary conditions. The total number of sites is $L = 10$.

$l_A = 3$. The BOMI in the chaotic phase, shown in (a), decreases instantly for PBC but has a delay for OBC and this delay is later for larger $s$. The BOMI then settles on a value close to zero for $l_B < L/2$. In the integrable phase, the BOMI for transvere field, shown in (b), shows unpredictable behaviour, but is negative and decreases in general. The BOMI for longitunal field in (c) shows the same behavior as the full overlap configuration in Figure 3(c). The TOMI in the partial overlap configuration is similar to the full overlap configuration. For the chaotic phase, however, we see that the decrease of the TOMI is delayed for OBC with increasing delay for larger $s$.

In Figure 12, the BOMI and TOMI for the disjoint configuration is shown. In this figure, $l_A = l_B = l = 3$. the distance between the rightmost of $A$ and the leftmost of $B$ is $d$. The BOMI and TOMI for both chaotic and integrable phases start at zero. The BOMI in the chaotic phase in (a) shows a small bump at an early time and then returns close to zero. The BOMI in the integrable phase increases slightly for transverse field (b) and remains zero for longitudinal field (c). The TOMI for the disjoint configuration is similar to the full and partial overlap configurations.

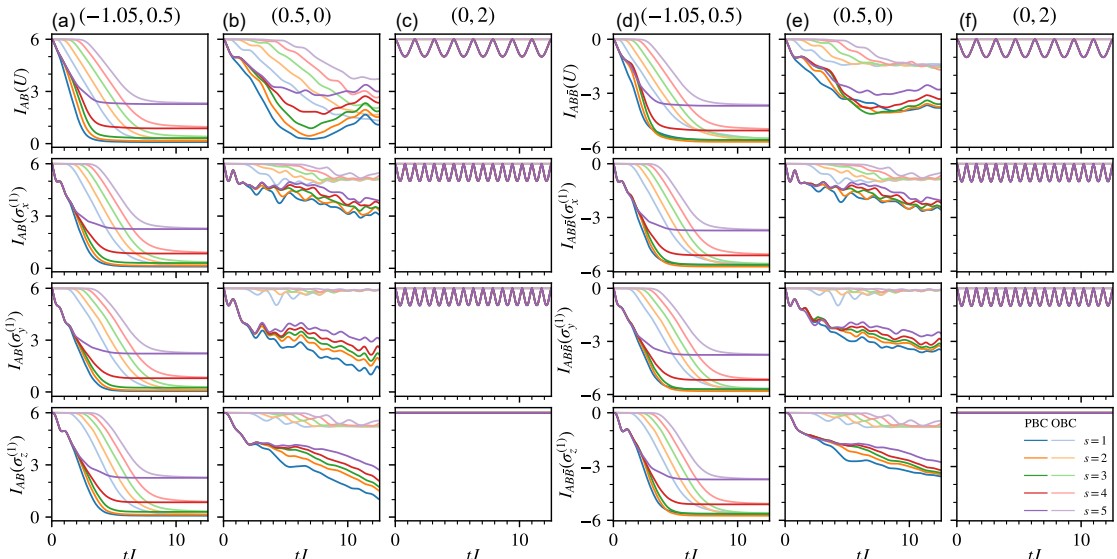

Figure 11: Time evolution of the (a-c) BOMI and (d-f) TOMI of the operators $U$, $\sigma_x^{(1)}$, $\sigma_y^{(1)}$, and $\sigma_z^{(1)}$ in the chaotic chain model for the partially overlapping configuration. The local operators are located at the left boundary of $A$ and $B$ and the size of $B$ is greater than the size of $A$ by $s$, $l_A + s = l_B$. The magnetic field is $(h_x, h_z) = (-1.05, 0.5)$ for (a,d), $(h_x, h_z) = (0.5, 0)$ for (b,e), and $(h_x, h_z) = (0, 2)$ for (c,f). Dark lines correspond to periodic boundary conditions and light lines correspond to open boundary conditions. The total number of sites is $L = 10$ and the number of sites in $A$ is $l_A = 3$.

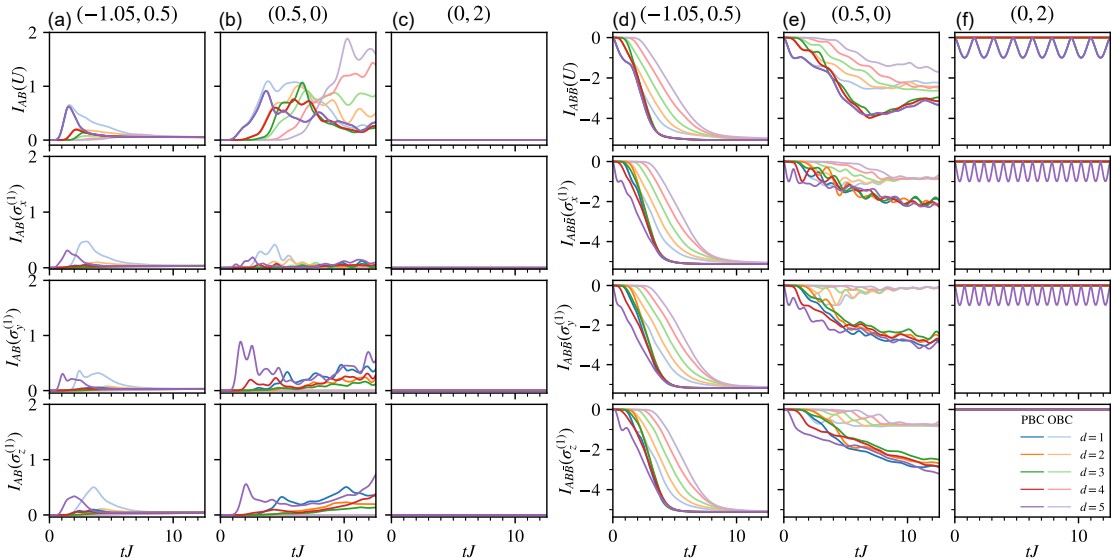

Figure 12: Time evolution of the (a-c) BOMI and (d-f) TOMI of the operators $U$, $\sigma_x^{(1)}$, $\sigma_y^{(1)}$, and $\sigma_z^{(1)}$ in the chaotic chain model for the disjoint configuration. The local operators are located at the left boundary of $A$ and the left boundary of $B$ is a distance $d$ from the right boundary of $A$. The magnetic field is $(h_x, h_z) = (-1.05, 0.5)$ for (a,d), $(h_x, h_z) = (0.5, 0)$ for (b,e), and $(h_x, h_z) = (0, 2)$ for (c,f). Dark lines correspond to periodic boundary conditions and light lines correspond to open boundary conditions. The total number of sites is $L = 10$ and the number of sites in $A$ and $B$ are $l_A = l_B = 3$.

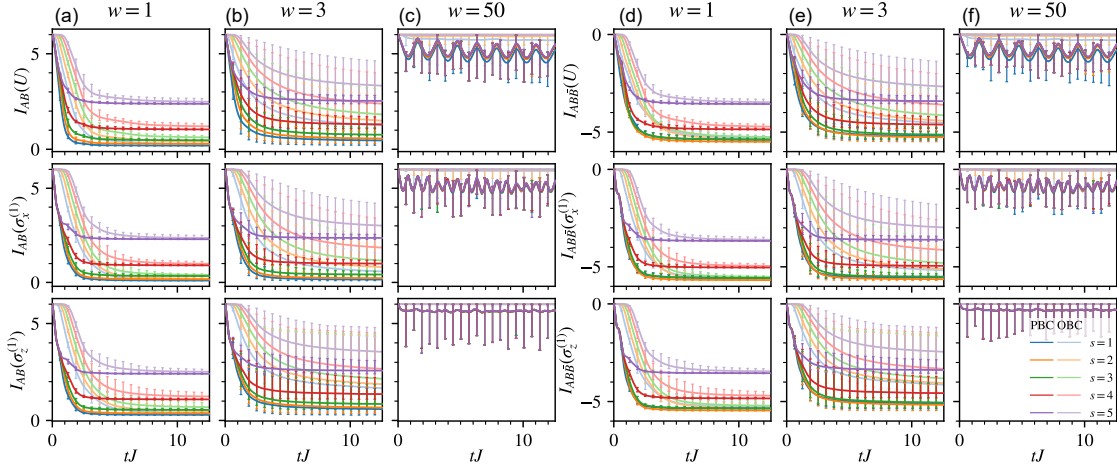

Figure 13: Time evolution of (a-c) BOMI and (d-f) TOMI of the operators $U$, $\sigma_x^{(1)}$, $\sigma_y^{(1)}$, and $\sigma_z^{(1)}$ in the disordered spin chain model for the partially overlapping configuration. The local operators are located at the left boundary of $A$ and $B$ and the size of $B$ is greater than the size of $A$ by $s$, $l_A + s = l_B$. The disorder strength is (a,d) $w = 1$, (b,e) $w = 3$, and (c,f) $w = 50$. Solid lines show the mean over 20 disorder configurations and error bars show the min and max. Dark lines correspond to periodic boundary conditions and light lines correspond to open boundary conditions. The total number of sites is $L = 10$ and the number of sites in $A$ is $l_A = 3$.

## B    The time evolution of BOMI and TOMI in the one-dimensional disordered Heisenberg model

Here, we will explain the time evolution of BOMI and TOMI in the partial overlap and disjoint configurations of the one-dimensional disordered Heisenberg model.

- **Small $w$ region ($w = 1$):** As in the full overlap configuration, the late time value of BOMI and TOMI in the partial overlap and disjoint configurations follows the Page's curve (See, Figures 13, and 14).

- **Near critical region ($w = 3$):** The time evolution of BOMI and TOMI depends on the operators and boundary conditions. In the partial overlap configurations, the late time value of BOMI and TOMI is larger than that in (34) with $w = 1$. In the disjoint configuration, the late time value of BOMI and TOMI in the chaotic phase ($w = 1$) is independent of the distance between two intervals, $d$, but that of BOMI and TOMI near the critical point ($w = 3$) depends on $d$. In (34) with OBC, the late time value of BOMI is a monotonically decreasing function of $d$, but that of TOMI is a monotonically increasing function of $d$. The $d$-dependence of BOMI shows MBL prevents the information from spreading, and that of TOMI shows MBL prevents the information from being delocalized.

- **Large $w$ region ($w = 50$):** As in Figure 13, for the partial overlap configuration, the value of BOMI and TOMI with the enough strong disorder increases, when $w$ increases. As in Figure 14, when $w$ increases, the value of the BOMI with the enough strong disorder in the disjoint configuration decreases, but the value of the TOMI decreases. In the disjoint configuration, the late time value of BOMI and TOMI in the chaotic phase ($w = 1$) is independent of the distance between two intervals, $d$, but that of BOMI and TOMI near the critical point ($w = 3$) depends on $d$. In (34) with OBC, the late time value of BOMI is a monotonically decreasing function of $d$, but that of TOMI is a monotonically

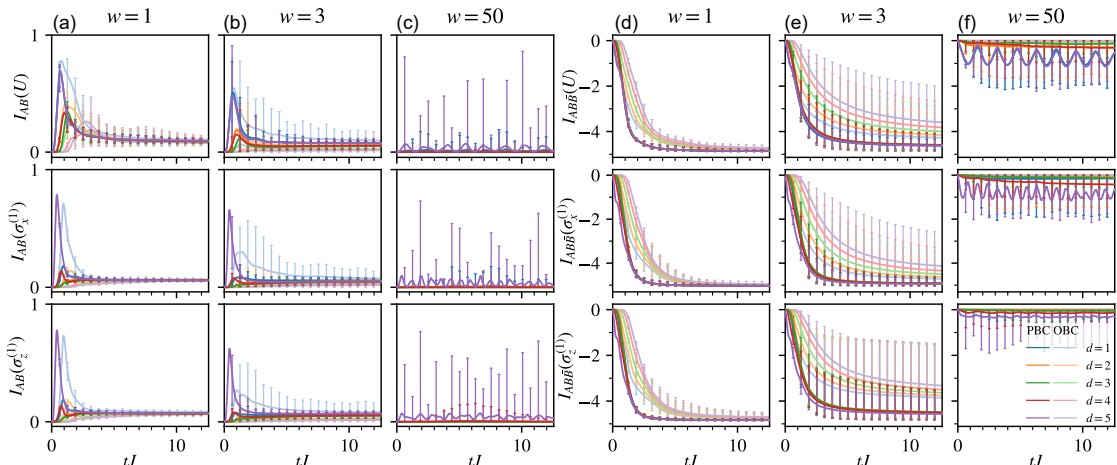

Figure 14: Time evolution of (a-c) BOMI and (d-f) TOMI of the operators $U$, $\sigma_x^{(1)}$, $\sigma_y^{(1)}$, and $\sigma_z^{(1)}$ in the disordered spin chain model for the disjoint configuration. The local operators are located at the left boundary of $A$ and the left boundary of $B$ is a distance $d$ from the right boundary of $A$. The disorder strength is (a,d) $w = 1$, (b,e) $w = 3$, and (c,f) $w = 50$. Solid lines show the mean over 20 disorder configurations and error bars show the min and max. Dark lines correspond to periodic boundary conditions and light lines correspond to open boundary conditions. The total number of sites is $L = 10$ and the number of sites in $A$ and $B$ are $l_A = l_B = 3$.

increasing function of $d$. The $d$-dependence of BOMI shows MBL prevents the information from spreading, and that of TOMI shows MBL prevents the information from being delocalized. As in Figure 14, when $w$ increases, the value of the BOMI with $w > w_{\text{cri}}$ in the disjoint configuration decreases, but the value of the TOMI decreases.

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
