# Peer review of "Local Operator Entanglement in Spin Chains"

_SciPost Physics Core, doi:SciPost Phys. Core 6, 070 (2023)_

## Round 3 · Referee Report · Anonymous (Referee 1) · 2021-4-30

Report

This paper presents numerical data for entanglement measures in a chaotic and a disordered chain, namely the mutual information and the “tripartite mutual information” for time-evolving operators. It is clear that considerable work has gone into the generation of the data and the presentation of this data and of background material. However, this manuscript does not appear to me to be publishable in Scipost. What is lacking is (a) a clearly defined motivation for studying these quantities and (b) clear lessons from the numerical data. The reader is shown many plots without necessarily knowing what to infer from them.

The authors do make attempts to interpret some of the findings (they describe a “light cone picture”, they use some basic properties of strongly entangled states, and they invoke the l-bit picture for many-body localization in the disordered case). However these theoretical accounts seem relatively limited. For example the discussion of the chaotic phase is not quantitative (except in the relatively simple t->∞ limit).

The observables considered are linear combinations of operator entanglement entropies. Therefore in the chaotic phase existing ideas can be used to give a more quantitative account of their behavior, to the extent that this is universal at large scales. The scaling theory for operator entanglement in chaotic chains of Ref 19 applies here.

Regardless of the presence or absence of a theoretical explanation, clearer physical motivation for studying these quantities is needed. If I understand correctly, the idea behind the tripartite information of a channel in previous work was to determine whether information that is initially encoded locally is encoded locally or globally after the application of the channel. In the present paper the context is different (local operators) and the motivation is less clear to me. It is true that the local operators studied here happen to be unitary, so they can be viewed as channels if desired, but the authors do not give an argument that it is useful to do this.

The paper is currently rather long. The authors might wish to decide what the key message(s) of their paper should be, and cut down the length and the number of figures in the main text to give the paper some focus.

Here are more specific comments.

Eq 1 - I understand the general idea the authors are trying to convey, but this equation is not correct and neither is the claimed relation between energy and temperature, in general.

Eq 3 - missing normalization

Eq 5 - can this be reformulated to be clearer?

Eq 9 - this quantity is claimed to be “exponentially large” - isn’t it bounded and of order 1 in the models studied here?

Sec 2.2. “we expect quantum information to be preserved locally”. Generically only classical information is preserved locally in the MBL phase (the z-component of the l-bit is conserved but not the x-component).

p6 “robust against noise” What is meant here? Noise (time-dependent fluctuations in the Hamiltonian) will generically destroy the MBL phase.

“linear in non-interacting systems or general non-integrable systems without localization”. This statement assumes translational invariance. Otherwise, sublinear growth is possible, even without localization.

p9 “unless O is the identity operator” This is not correct, there are other O with this property

p10, 11 contain many vague (e.g. “unrelated to each other”, “data regarding the local entanglement structure”) or repetitive statements

p19 should S_A,B be I_A,B?
  • validity: -
  • significance: -
  • originality: -
  • clarity: -
  • formatting: -
  • grammar: -

Author:  Eric Mascot  on 2023-08-25  [id 3926]

(in reply to Report 1 on 2021-04-30)
Category:
answer to question
correction

We would like to express our sincere gratitude for taking the time to read our paper and for providing us with your invaluable feedback. We greatly appreciate your numerous insightful comments. We have addressed the key points, (a) providing a clear motivation, and (b) enhancing the clarity and comprehensibility of our analysis of the numerical data. To address the first point, we have incorporated an additional paragraph at the end of the introduction, which explicitly outlines the motivation behind our investigation of BOMI and TOMI. Regarding the second point, we have reworked the summary on page 3 to better articulate the results obtained in our paper. Furthermore, we have diligently reviewed your suggestions and made substantial revisions throughout the entire paper. Here, we briefly explain the motivation and analysis.

・Motivation for the Ising chain with magnetic field: We would like to study whether the late time evolution of the BOMI and TOMI of local operators depends on the operators in systems with the strong scrambling ability and few degrees of freedom. In two-dimensional holographic CFT, systems with the strong scrambling effect and large degrees of freedom have a late time value of BOMI and TOMI that does not depend on the local operator. Since the initial state considered in this paper is characterized by the Pauli operators, the fact that the late time BOMI and TOMI are independent of the local operator suggests that the scrambling effects washes out the information about the initial state, and the steady state is typical. Additionally, we would like to study how finite system size affects the scrambling ability of the dynamics. ・What we obtained: We found that regardless of the operators and boundary conditions, the late time value of BOMI and TOMI follows Page’s curve. This suggests the scrambling effect washes out the information about the initial state and boundary conditions, so that the late time steady state is typical. We also found that the late time value of TOMI and BOMI can depend on the system size. This suggests that the finiteness of the system prevents the dynamics from breaking non-local correlation. ・Motivation for the disordered Heisenberg chain: We would like to study the properties of circuits with weak scrambling effects that preserve the quantum nature of the initial state. As a remarkable example of a circuit that would have such a weak scrambling effect, we considered a spin system with a chaotic and MBL phase. ・What we obtained: In regions of weak disorder strength, regardless of boundary conditions and local operators, we find that the late time value of TOMI and BOMI follows the Page curve, as in Ising model in the chaotic region. In the regions of strong disorder strength, TOMI and BOMI periodically evolve in time. The period is determined by the local interaction. We propose an effective model describing the time evolution of BOMI and TOMI in the Heisenberg model with the weak disorder. In this model, the periodic motion of BOMI and TOMI is caused by the off-diagonal terms when the density matrix is expanded in the z-directional basis. Thus, this periodic motion may be due to the quantum nature of the state. Another suggestion was that the paper is rather long and should be shortened a bit. We moved many results to the appendices. However, after adding explanations for clarity, the manuscript has become a bit longer than the previous version. If the paper is too lengthy, we will shorten it and would appreciate it if you could point this out to us.

In the following, we respond to your individual questions and suggestions. We hope you will be satisfied with our responses.

  • Eq 1 - I understand the general idea the authors are trying to convey, but this equation is not correct and neither is the claimed relation between energy and temperature, in general.

Thank you for a useful comment. We have removed the last approximate expression in Eq. (1) as we think it may lead to confusion and was not necessary for the subsequent discussion. To prevent any confusion for the reader, we have also rewritten Eq. (3) to define thermalization of a subsystem.

  • Eq 3 - missing normalization

Thank you for pointing this out. We have added the normalization constant to Eq. (3).

  • Eq 5 - can this be reformulated to be clearer?

Thank you for your comment. To make the explanation clearer, we have added the following sentences: “If the late time behavior of observables in V are approximated by their thermal expectation value, then we say that thermalization of V occurs. For example, in addition to the mutual information content studied in this paper, the one-point function and the distance between states are also used as indicators of thermalization of subsystems. If the sybsystem thermalizes, these observables behave as follows:” We have also changed Eqs. (4,5).

  • Eq 9 - this quantity is claimed to be “exponentially large” - isn’t it bounded and of order 1 in the models studied here?

Thank you for pointing this out and apologize for the confusion. The part you pointed out was written in such a way that the results from the two-dimensional holographic CFT would be confused with the results from the spin system. Therefore, we have rewritten the description to first refer to the behavior of the squared commutation relation in the two-dimensional holographic CFT, and then refer to the behavior in the spin system.

  • Sec 2.2. “we expect quantum information to be preserved locally”. Generically only classical information is preserved locally in the MBL phase (the z-component of the l-bit is conserved but not the x-component).

We agree with the referee and remove this misleading paragraph.

  • p6 “robust against noise” What is meant here? Noise (time-dependent fluctuations in the Hamiltonian) will generically destroy the MBL phase.

Thank you for your question. We agree with the referee that time-dependent fluctuations in the Hamiltonian will generically destroy the MBL phase. We have changed the sentences to clearly distinguish between spatial and temporal fluctuations: “Also, note that systems with spatially quasiperiodic modulation, whose non-uniform feature is determined by a few parameters as opposed to lacking long-range correlation, can be many-body localized. Even in these cases, we can generally expect the many-body localized phase to be robust against time-independent weak perturbations.”

  • “linear in non-interacting systems or general non-integrable systems without localization”. This statement assumes translational invariance. Otherwise, sublinear growth is possible, even without localization.

We agree with the referee and add “translationally invariant” between “general” and “non-integrable”.

  • p9 “unless O is the identity operator” This is not correct, there are other O with this property

Thank you for pointing this out. We have rewritten the part as follows: For example, if O(x, t) is the identity operator, then the entanglement structure of the state in (20) is the same as that of maximally entangled state. In the general case, O(x, t) can make the quantum entanglement structure of the state in (20) less entangled than that of the maximally entangled state.

  • p10, 11 contain many vague (e.g. “unrelated to each other”, “data regarding the local entanglement structure”) or repetitive statements

Thank you for the comment. We have added Section 2.3.2 and sentences under Eq. 25 on page 11 to clarify the explanation. Also, we have revised the explanation in Page 12 and 13. In Section 2.3.2, we explain that the initial state is given by the direct product state of EPR pairs. Under Eq. 25 on page 11, we explain that BOMI and TOMI are quantities that measure how many of these EPR pairs are lost locally. In Page 12 and 13, we first explain why we study BOMI and TOMI in the case of three configurations: full overlap, partial overlap, and disjoint configuration. The reason is as follows. In 2D CFTs, the effect of scrambling on BOMI and TOMI depended greatly on the way the configuration is set up, so in the spin system we expect that the effect of scrambling on BOMI and TOMI would also depend greatly on the way the configuration is set up. In addition, we have explained how the time evolution of BOMI and TOMI in each configuration can be interpreted in terms of the EPR pairs. 

  • p19 should S_A,B be I_A,B?

Thank you for the comment. As you said, it should be $I_{A,B}$. Therefore, we have changed to the following: Then, the behavior of $I_{A,B}$ following the plot of Figure 7 is consistent with (30) with $\hat{L} = 2l + s$.

We sincerely hope that these revisions meet your expectations and address the concerns you raised. We are grateful for the opportunity to enhance the clarity and impact of our paper through your valuable input. Once again, we would like to express our appreciation for your time and expertise in reviewing our work.

---

## Round 3 · Referee Report · Anonymous (Referee 2) · 2021-5-12

Report

The authors study the dynamics of bipartite and tripartite operator entanglement entropy of one dimensional spin-1/2 models. These measures have been studied for clean chaotic and integrable cases, along with chaotic and MBL phases of disordered spin chains. These multipartite measures can provide a more granular understanding of these well-known phases of matter. The main conclusion of the work is characterising the steady state in these cases, in particular finding evidence for the Page curve of a random pure state in the clean chaotic regime.

Although the topic is of interest in the context of operators spreading, in my opinion the current version of the manuscript hasn't made significant progress to warrant publication in its current form. It is unclear in what sense the results go beyond the existing knowledge about these phenomena. It is likely that some new insights can be gleaned from the numerical data, but in the absence of significant theoretical analysis the impact of the numerics on its own is inconclusive.

I suggest a few theoretical questions which could sharpen the results of the paper.

  1. Do we expect change in behaviour in TOMI in non-integrable and integrable cases? If yes, why?

  2. Does the Page curve apply also in the integrable and MBL cases? The steady states have volume law entanglement in both cases? How should we understand this theoretically?

  3. MBL is associated with logarithmic growth of von Neumann entanglement entropy? How does that effect the results on BOMI and TOMI?

As a practical point, the error bars for the data in the disordered case are quite large. In its current form it is hard to draw any meaningful conclusion from the data. I expect with more disorder averaging the trends will be much clearer.

In its present form, the paper includes a lot of data and plots but not as much physical interpretation. It may even be feasible to reduce the number of figures in order to provide a sharper focus on the main physical consequences of the numerical results.

  • validity: -
  • significance: -
  • originality: -
  • clarity: -
  • formatting: -
  • grammar: -

Author:  Eric Mascot  on 2023-08-25  [id 3925]

(in reply to Report 2 on 2021-05-12)
Category:
answer to question
correction

We would like to thank you for reviewing the paper and providing valuable feedback. We have taken great care to address the “absence of significant theoretical analysis” by thoroughly revising the text to elucidate the significance of our results in a clear and understandable manner. We have made significant changes to the “Summary of results” sections which outlines our findings. Here, we briefly explain the motivation and analysis.

・Motivation for the Ising chain with magnetic field: We would like to study whether the late time evolution of the BOMI and TOMI of local operators depends on the operators in systems with the strong scrambling ability and few degrees of freedom. In two-dimensional holographic CFT, systems with the strong scrambling effect and large degrees of freedom have a late time value of BOMI and TOMI that does not depend on the local operator. Since the initial state considered in this paper is characterized by the Pauli operators, the fact that the late time BOMI and TOMI are independent of the local operator suggests that the scrambling effects washes out the information about the initial state, and the steady state is typical. Additionally, we would like to study how finite system size affects the scrambling ability of the dynamics. ・What we obtained: We found that regardless of the operators and boundary conditions, the late time value of BOMI and TOMI follows Page’s curve. This suggests the scrambling effect washes out the information about the initial state and boundary conditions, so that the late time steady state is typical. We also found that the late time value of TOMI and BOMI can depend on the system size. This suggests that the finiteness of the system prevents the dynamics from breaking non-local correlation. ・Motivation for the disordered Heisenberg chain: We would like to study the properties of circuits with weak scrambling effects that preserve the quantum nature of the initial state. As a remarkable example of a circuit that would have such a weak scrambling effect, we considered a spin system with a chaotic and MBL phase. ・What we obtained: In regions of weak disorder strength, regardless of boundary conditions and local operators, we find that the late time value of TOMI and BOMI follows the Page curve, as in Ising model in the chaotic region. In the regions of strong disorder strength, TOMI and BOMI periodically evolve in time. The period is determined by the local interaction. We propose an effective model describing the time evolution of BOMI and TOMI in the Heisenberg model with the weak disorder. In this model, the periodic motion of BOMI and TOMI is caused by the off-diagonal terms when the density matrix is expanded in the z-directional basis. Thus, this periodic motion may be due to the quantum nature of the state.

In the following, we respond to your individual questions and suggestions. We hope you will be satisfied with our responses.

  • Do we expect change in behaviour in TOMI in non-integrable and integrable cases? If yes, why?

We thank you for your question. Yes, the behavior of TOMI depends on the integrability. More precisely, the behavior depends on the scrambling effects of the dynamics (e.g., arXiv:1511.04021, 1812.00013). The time-evolved states in this paper are initially equivalent to a product of EPR pairs. BOMI is proportional to the number of EPR pairs in the union of A and B. If dynamics do not scramble, this EPR pair picture works well, and the BOMI is zero. Conversely, if the dynamics do have scrambling ability, some of the EPR pairs are locally hidden in the time evolution, so that TOMI becomes negative. A large negative value of the TOMI suggests a strong scrambling ability of the dynamics. We have added this explanation below Eq. 25.

  • Does the Page curve apply also in the integrable and MBL cases? The steady states have volume law entanglement in both cases? How should we understand this theoretically?

In the chaotic regime, the strong scrambling washes out the information about the local operator or boundary conditions. In this case, we expect the entanglement entropy to follow the Page curve. We have added a paragraph in section 3.2 describing the theoretical understanding.

  • MBL is associated with logarithmic growth of von Neumann entanglement entropy? How does that effect the results on BOMI and TOMI?

Operator entanglement entropy does not appear to logarithmically increase in time. Generally, entanglement entropy strongly depends on the initial state, subsystem, and the time evolution operator. When entanglement entropy grows logarithmically in time, the initial state is in an unentangled state, and the subsystem is taken to be half of the total system, as in PRL 109, 017202 and PRB 104, 214202. In this paper, the initial state is given by a product of EPR states, which we have added in section 2.3.2, and the subsystem is not half of the total system. Therefore, the OEE does not grow logarithmically in time. Instead, BOMI and TOMI in MBL is periodic in time (quantum revival). The periodic motion is determined by the strength of the disorder, as described in section 4. This periodicity suggests the state returns to the initial state, and the time evolution operator in MBL may preserve the quantum nature of the state.

  • As a practical point, the error bars for the data in the disordered case are quite large. In its current form it is hard to draw any meaningful conclusion from the data. I expect with more disorder averaging the trends will be much clearer.

We apologize for the confusion caused by the “error” bars. These bars show the min-max range over disorder configurations. We believe the disorder averaged lines show clear trends without the need for additional averaging.

  • In its present form, the paper includes a lot of data and plots but not as much physical interpretation. It may even be feasible to reduce the number of figures in order to provide a sharper focus on the main physical consequences of the numerical results.

Thank you for the useful feedback. We have moved many of the figures to the appendix. We have added a section describing each configuration shown in Figure 2. In 2D CFT, the effect of scrambling on the time evolution of BOMI and TOMI is highly dependent on the configuration. However, in the spin systems considered in this paper, the full overlap configuration shows a dependence on the scrambling ability. Therefore, we move the partial overlap and disjoint configurations to the appendix.

---

## Round 5 · Referee Report · Anonymous · 2023-9-7

Report
The authors have made efforts to address the comments. The essential structure of the paper remains the same and still feels a little unbalanced, with more qualitative discussion and general review (e.g. of MBL) than may be necessary (although the authors have made some effort to lighten the manuscript by taking some of the results out of the main text). It remains true that many of the take-aways from the numerical results are rather qualitative, and it is not clear to me that they go beyond previous theoretical analysis such as the reference by Kudler-Flam et al. Nevertheless the numerical results may be a reference point for future works. The background material may also be useful for some readers. On these bases the paper may now be publishable in Core.
Anonymous on 2023-09-08 [id 3964]
The paper investigates the operator entanglement of the time-evolution operator for chaotic, integrable and many-body localized systems. It focuses on bipartite and tripartite operator mutual information to quantify the entangling properties of the time evolution. By mapping the density matrix to a channel and using that to numerically evaluate the mutual information, the authors are able to describe some of the universal features of the dynamics. For this they consider the clean spin-1/2 Ising model in its integrable and non-integrable limits, and the random-field Heisenberg model, for studying many-body localization (MBL). They have investigated the early and late behaviour of these observables in the different dynamical regimes.
Their findings suggest that in the clean case that the short time behaviour is consistent with a light cone and at late times the state fits the Page curve. They draw on similarities with 2D holographic CFTs to explain their results. While this would be reasonable, but the field theory may not apply to spin chain in question. There is a whole body of literature studying chaos and light cone dynamics in spin chains. An attempt should be made to connect the results with the known results in the literature. In the integrable limit, the quantities do not decay with time. Can this be understood analytically even from the perspective of free-field theory? The connection to free field theory can be made more thorough. In this case as well, there is a large body of work studying entanglement dynamics in integrable spin chains, and connecting their results with the literature would be quite appropriate.
In the disordered model, at weak disorder the results are consistent with chaotic dynamics. While at strong disorder in the localized phase the oscillations can be described by a simple effective Ising model. The number of disorder realizations (20) is too low for this analysis and therefore the error bars are large. It would be reasonable to average over more disorder realizations.
In both the cases (clean and disordered) some scaling with system (L=12, 14) can be attempted. It should not be computationally prohibitively expensive.
Overall, the manuscript has some new results which provide additional understanding of chaotic and many-body localized systems in terms of operator entanglement. There are three main points to address 1. Discussion relating to known results on entanglement dynamics in chaotic and integrable spin chains. 2. Finite size scaling of the results. 3. Reduce error bars by using a larger number of disorder realizations.
With these changes I can recommend the manuscript for publication.

---

## Round 5 · Author Response

Dear editor-in-charge,
I am writing to resubmit the revised version of our manuscript titled "Local Operator Entanglement in Spin Chains”. We are grateful for the thoughtful feedback provided by the referees, and we believe that the revisions we have made address the issues raised comprehensively.
We have carefully revised the manuscript according to the reviewer's feedback, and we have also made additional improvements to enhance the clarity, analysis, and overall contribution of the work. Specifically, we have made the following changes:
-
We have restructured the manuscript to focus on the “full overlap” configuration and moved other configurations to the appendix.
-
We have included additional analysis and discussion.
-
We have carefully proofread the manuscript to eliminate any grammatical errors or typographical mistakes.
We believe that these revisions align the manuscript much more closely with the scope and standards of SciPost Physics Core, and we are confident that the updated version represents a substantial contribution to the field. We appreciate the opportunity to resubmit our work to SciPost Physics Core and are hopeful for a positive outcome.
Sincerely, Eric Mascot Masahiro Nozaki Masaki Tezuka

---

## Round 5 · List of Changes

- Revised the first few sentences of the abstract
- Revised the introduction to provide clear motivation and to better articulate the results
- Provided additional analysis of the data
- Improved the description of thermalization
- Improved the description of OTOC
- Clarified "robust against noise" for the MBL phase
- Corrected operators with maximal entanglement structure
- Added a discussion of EPR pairs in a subsystem as an interpretation of BOMI and TOMI
- Added a discussion of factorization of the density matrix as an interpretation of late-time BOMI
- Moved "partial overlap" and "disjoint" configurations to the appendix
- Added a discussion on the effective model for large disorder
- Many minor corrections

You are currently on this page

---

## Editorial Decision

published